# Subterranean mammals show convergent regression in ocular genes and enhancers, along with adaptation to tunneling

Raghavendran Partha[1], Bharesh K Chauhan[2,3], Zelia Ferreira[1], Joseph D Robinson[4], Kira Lathrop[2,3], Ken K Nischal[2,3], Maria Chikina[1]*, Nathan L Clark[1]*

[1]Department of Computational and Systems Biology, University of Pittsburgh, Pittsburgh, United States; [2]UPMC Eye Center, Children's Hospital of Pittsburgh, Pittsburgh, United States; [3]Department of Ophthalmology, University of Pittsburgh School of Medicine, Pittsburgh, United States; [4]Department of Molecular and Cell Biology, University of California, Berkeley, United States

**Abstract** The underground environment imposes unique demands on life that have led subterranean species to evolve specialized traits, many of which evolved convergently. We studied convergence in evolutionary rate in subterranean mammals in order to associate phenotypic evolution with specific genetic regions. We identified a strong excess of vision- and skin-related genes that changed at accelerated rates in the subterranean environment due to relaxed constraint and adaptive evolution. We also demonstrate that ocular-specific transcriptional enhancers were convergently accelerated, whereas enhancers active outside the eye were not. Furthermore, several uncharacterized genes and regulatory sequences demonstrated convergence and thus constitute novel candidate sequences for congenital ocular disorders. The strong evidence of convergence in these species indicates that evolution in this environment is recurrent and predictable and can be used to gain insights into phenotype–genotype relationships.
DOI: https://doi.org/10.7554/eLife.25884.001

*For correspondence:
mchikina@pitt.edu (MC);
nclark@pitt.edu (NLC)

**Competing interests:** The authors declare that no competing interests exist.

## Introduction

The subterranean habitat has been colonized by numerous animal species for its shelter and unique sources of food (*Andersen, 1987*; *Nevo, 1979*). Obligate fossorial species in particular have adopted the underground as a dedicated home, yet the intense demands of life underground often require unique specializations. For one, the air present in tunnels is often low in oxygen (hypoxic) and high in carbon dioxide (hypercapnic) (*Nevo, 1979*). This dark environment also requires the development of enhanced senses to compensate for loss of vision. These and other subterranean specializations have been reported in many independent evolutionary lineages of insects, amphibians, reptiles, and mammals (*Leys et al., 2003*; *Lacey et al., 2000*; *Albert et al., 2007*; *Wilkinson, 2012*). Within mammals alone, there are several unrelated subterranean species, including the true moles (family Talpidae), the African golden moles (Chrysochloridae), and the marsupial moles (Notoryctidae). There are also at least three unrelated lineages of subterranean rodents: the naked mole-rat (*Heterocephalus glaber*), blind mole-rats (Spalacidae), and the pocket gophers (Geomyidae).

Fossorial mammals have evolved a number of morphological and physiological traits that are considered adaptations to subterranean life, affecting how they sense their environment, how they move through it, or how they deal with its physiological demands. For one, species that dig with their forelimbs, like the star-nosed mole, have enlarged forelimbs and claws that allow them to

**eLife digest** Over the past 100 million years, many mammals, such as moles or mole rats, for example, have evolved to live almost entirely underground. During their transition to adapt to life underground, many species have reduced or completely lost their sense of sight, and often have only a small remnant of an eye that can sometimes be completely covered by skin and fur.

In addition, the sections of the DNA that usually control how the eyes form have changed in these animals. Since there is less need for a working eye in dark environments, DNA related to the eye is no longer protected from damaging mutations in mammals that live underground. So, by comparing the DNA of mammals that live aboveground and underground, scientists can identify the parts of DNA that help form mammals' eyes.

Previous studies have discovered many sections of DNA responsible for producing the proteins that make up the eye. However, scientists know less about which sections of DNA control when and where these proteins are made. To address this, Partha et al. have studied the DNA of four underground mammals: the star-nosed mole, the cape golden mole, the naked mole-rat and the blind mole-rat.

By comparing the DNA of these animals with that of mammals that live above ground, Partha et al. identified sections of DNA that contained an abnormally high number of changes in the blind underground mammals. Many of these sections are involved in forming the eye, including controlling when and where proteins are made. Overall, the findings show that comparing rates of evolution in different species can help uncover sections of DNA that guide and influence how organisms develop.

Understanding how the eye is formed is not only of interest to scientists studying evolution and biology; it also has wider applications in healthcare. Many people suffer from unexplained eye abnormalities, and insight into the sections of DNA that control the eye's development could help medical professionals diagnose these cases and design new treatments.
DOI: https://doi.org/10.7554/eLife.25884.002

tunnel through substrate, whereas species that dig with their teeth, such as the naked mole-rat, have reduced limbs and continuously growing incisors (*Dubost, 1968*; *Ellerman, 1956*). In the absence of light, non-visual sensory systems have been elaborated. These include the sensitive vibrissae (i.e., sensory hairs) of the naked mole-rat and the large and elaborate snout of the star-nosed mole, which has become a remarkable 'tactile eye' (*Eloff, 1951*; *Hill et al., 2009*; *Catania, 1999*). Fossorial mammals have also evolved an ability to withstand hypoxic conditions that rivals that of species living at high altitude (*Nevo, 1979*; *Ar et al., 1977*). For example, some species have high erythrocyte counts and high skeletal-muscle myoglobin to facilitate oxygen exchange. Some of these adaptive strategies are shared by different fossorial lineages and as such represent prime examples of convergent evolution (*Nevo, 1979*). Yet, some of the most striking convergent transformations in subterranean mammals are the reductions and losses of traits shared among aboveground mammals, of which the most prominent example is the reduction of the eye (*Dubost, 1968*; *Hill et al., 2009*; *Cei, 1946a*, *1946b*).

Vision in many subterranean mammals is limited, and the degree of limitation in each species is related to the extent of its underground habitation (*Nemec et al., 2008*; *Quilliam, 2009*; *Sanyal et al., 1990*). For example, star-nosed moles (*Condylura cristata*) that share their time aboveground and underground possess diminutive eyes with thick eyelids (*Catania, 1999*), whereas the naked mole-rat , which spends almost all of its time underground, has tiny eyes that are rarely opened (*Hetling et al., 2005*). Even more extreme are the completely subcutaneous eyes of the cape golden mole (*Chrysochloris asiatica*) and the blind mole-rat (genus *Nannospalax*), which are thought to reflect their strictly subterranean lifestyle (*Sanyal et al., 1990*; *Sweet, 1909*). While some degree of visual regression is shared between subterranean mammals, not all visual structures and genetic pathways have regressed to the same degree. For instance, the eyes of true moles and mole-rats show anatomical regression and always exhibit a small eye but retain ocular architecture, suggesting that the basic eye developmental programs must be largely intact in these animals (*Carmona et al., 2008, 2010*; *Quilliam, 1966*). The convergent loss of vision and visual structures in

subterranean mammals allows us to ask which genetic regions – coding or non-coding – contributed to regression in these species and which were conserved.

The genetic causes of these malformations have been probed through studies of blind cavefish and evolutionary analysis of retinal genes in subterranean mammals (*Jeffery, 2009*; *Emerling and Springer, 2014*). Pioneering work by Hendriks et al. found the evolutionary rate of the lens and retina protein αA-crystallin to be markedly accelerated in the Middle Eastern blind mole-rat (*Spalax ehrenbergi*), as would be expected under relaxed constraint (*Hendriks et al., 1987*). Furthermore, *Emerling and Springer (2014)* revealed that regressive genetic changes in retinal proteins are unevenly distributed across different visual pathways and eye tissues. Previous studies have placed more emphasis on retinal components of vision and connections to the visual cortex because it is these components that sense light and transmit images to the brain for vision (*Emerling and Springer, 2014*; *Cooper et al., 1993*). Less emphasis has been placed on the genes contributing to other eye tissues, such as the cornea.

The genomes of four subterranean mammals have been sequenced and studied for changes that have occurred in response to their unique environment. The naked mole-rat genome revealed genetic changes in key genes involved in thermogenesis and circadian rhythm, as well as gene loss and deactivating mutations in core visual perception genes (*Kim et al., 2011*). The genome of the blind mole-rat (*Nannospalax gailili*) also yielded diverse insights into its subterranean adaptations, such as an impactful change to the P53 protein that allows cells to escape hypoxia-induced apoptosis, as well as the upregulation of specific pathways involved in the response to hypoxia and hypercapnia (*Fang et al., 2014*). Additionally, parallel evolution was seen in the deactivation of visual perception genes in the blind mole-rat and naked mole-rat. The convergence of such changes provides evidence that they have occurred in response to the subterranean environment rather than as a result of unrelated species-specific conditions or neutral processes, highlighting a potential strategy to discover additional genetic regions showing a similar response (*Losos, 2011*; *Stern, 2013*; *Rosenblum et al., 2014*).

Previous studies have used convergent evolution to reveal genetic changes that are related to environmental shifts without *a priori* expectations of which regions might respond. One strategy has been to search for convergent amino acid substitutions at specific protein sites (*Foote et al., 2015*; *Liu et al., 2010*; *Dobler et al., 2012*). A complementary strategy is to search for convergent changes in selective pressure on larger functional regions, such as genes or regulatory sequences, because evolution at different nucleotides within a gene could nevertheless lead to convergent phenotypic effects. In practice, convergent changes in selective pressure are inferred by studying evolutionary rates, because selective constraint slows evolution, whereas lack of constraint and adaptation speed it. Computational methods employing this strategy search for functional elements whose evolutionary rates changed on those branches exhibiting the convergent environmental change (*Marcovitz et al., 2016*; *Hiller et al., 2012*; *Chikina et al., 2016*; *Lartillot and Poujol, 2011*). One demonstration of this approach by our group identified genes that convergently responded when mammalian lineages shifted from a terrestrial to a marine environment (*Chikina et al., 2016*). Another recent study by *Prudent et al.(2016)* demonstrated that regions showing convergent rate acceleration in the subterranean environment were enriched in visual perception genes and also contained circadian rhythm genes. Together, these studies show the promise of convergent rates to reveal genes underlying major changes in morphology and physiology that are related to drastic environmental shifts.

To investigate the demands placed upon subterranean species by their extreme environment, we searched for genes exhibiting convergent rate changes in four subterranean mammals. We report a large set of genes showing marked relaxation of constraint in subterranean species, which were highly enriched for visual functions. This set also contained many genes of undetermined function, which could be unrecognized causative genes in eye-related diseases. Finally, we pinpointed the eye-specific transcriptional enhancers in the *Pax6* gene region using a new variant of our method and demonstrated the potential to detect new eye-specific enhancers at key developmental genes.

## Results

### Many genes have altered evolutionary rates specifically in subterranean mammals

We first sought to identify the genes that responded to conditions in the subterranean environment. Accordingly, we used relative evolutionary rate (RER) methods to identify protein-coding genes that evolved at a more rapid rate specifically on subterranean branches of the mammalian phylogenetic tree. Subterranean branches consisted of those leading to the star-nosed mole (*Condylura cristata*), the cape golden mole (*Chrysochloris asiatica*), the naked mole-rat (*Heterocephalus glaber*) and the blind mole-rat (*Nannospalax galili*). Each of these species represents a lineage that independently colonized the subterranean habitat, as each is more closely related to aboveground mammals than they are to each other (*Figure 1A*). Hence, similar phenotypic changes within these species are regarded as convergent traits. To demonstrate our RER methods, we first present the case of the eye-specific gene *LIM2*, which encodes Lens intrinsic membrane protein 2. First, the amount of amino acid divergence in LIM2 on each mammalian branch was quantified using sequences from 39 species and standard evolutionary models (*Figure 1B*) (see Materials and methods). The resulting LIM2 tree is markedly different from the genome-wide average tree in *Figure 1A*, and reveals distinctly high amounts of divergence in LIM2 for the four subterranean species. This rapid divergence probably resulted from loss of selective constraint in the dark subterranean environment. To quantify this rate acceleration in the LIM2 tree, we normalized all branch lengths for the expected amount of change as defined by the genome-wide average divergence for each branch. This average, after scaling (see Materials and methods), should reflect both the underlying speciation times in the mammalian phylogeny as well as changes in demographic factors affecting substitution rates. The resulting RER values for each branch are plotted in *Figure 1C*. An RER of zero indicates that LIM2 evolved at exactly the expected rate on that branch, while positive and negative values reflect faster and slower rates, respectively. By examining RERs it becomes clear that LIM2 changed at abnormally rapid rates in the four subterranean mammals; the rates for all four subterranean species are more rapid than all aboveground species, and this difference is supported statistically ($p=0.00084$, Mann-Whitney U test). Thus, extending the RER calculations to all other genes, we can distinguish the functions that responded during adaptation to the subterranean environment. Importantly, the convergence of these species allows us to confidently infer genes that responded specifically to subterranean life, because faster rates in all four species are not likely to be due to random fluctuations, as reflected by the low *P*-value for LIM2.

We performed the same RER analysis on 18,980 protein-coding genes to determine which shifted to faster or slower evolutionary rates specifically in subterranean species. We will hereafter refer to such genes as 'mole-accelerated' and 'mole-decelerated', respectively (see Materials and methods). At a false discovery rate (FDR) of 15%, we identified 55 mole-accelerated genes. We expect mole-accelerated genes to result from either selection for amino acid changes (i.e., positive Darwinian selection) or, alternatively, from a reduction in purifying selection, as suggested for the LIM2 protein. At the other extreme, we identified 1306 mole-decelerated genes at the same FDR. We expect genes to show rate deceleration if there is stronger purifying selection on that gene's function in the subterranean environment, perhaps as the result of increased importance for fitness.

### Vision-related functions are highly enriched among mole-accelerated genes

Genes with the strongest evidence of mole-acceleration were consistently associated with function in two organs, eye and skin. To illustrate, 17 of the top 30 mole-accelerated genes are expressed solely in eye tissues or are associated with eye-related disorders, whereas three accelerated genes are associated with skin, hair, and nails (*Table 1*). Among the genes showing very strong signals of mole-acceleration, we find proteins tha are specifically expressed in tissues of the eye such as the retina-specific proteins ROM1 and GNAT1 (*Figure 2*). The complete list of the 55 mole-accelerated genes similarly contains a large proportion that are related to vision and external tissues (*Supplementary file 1*), and they were highly enriched for functional annotations including eye morphology, photoreceptors, visual signal transduction, and eye-related mutant phenotypes (*Table 2*, *Supplementary file 2*). The strength of this enrichment is clearly illustrated by examining all genes

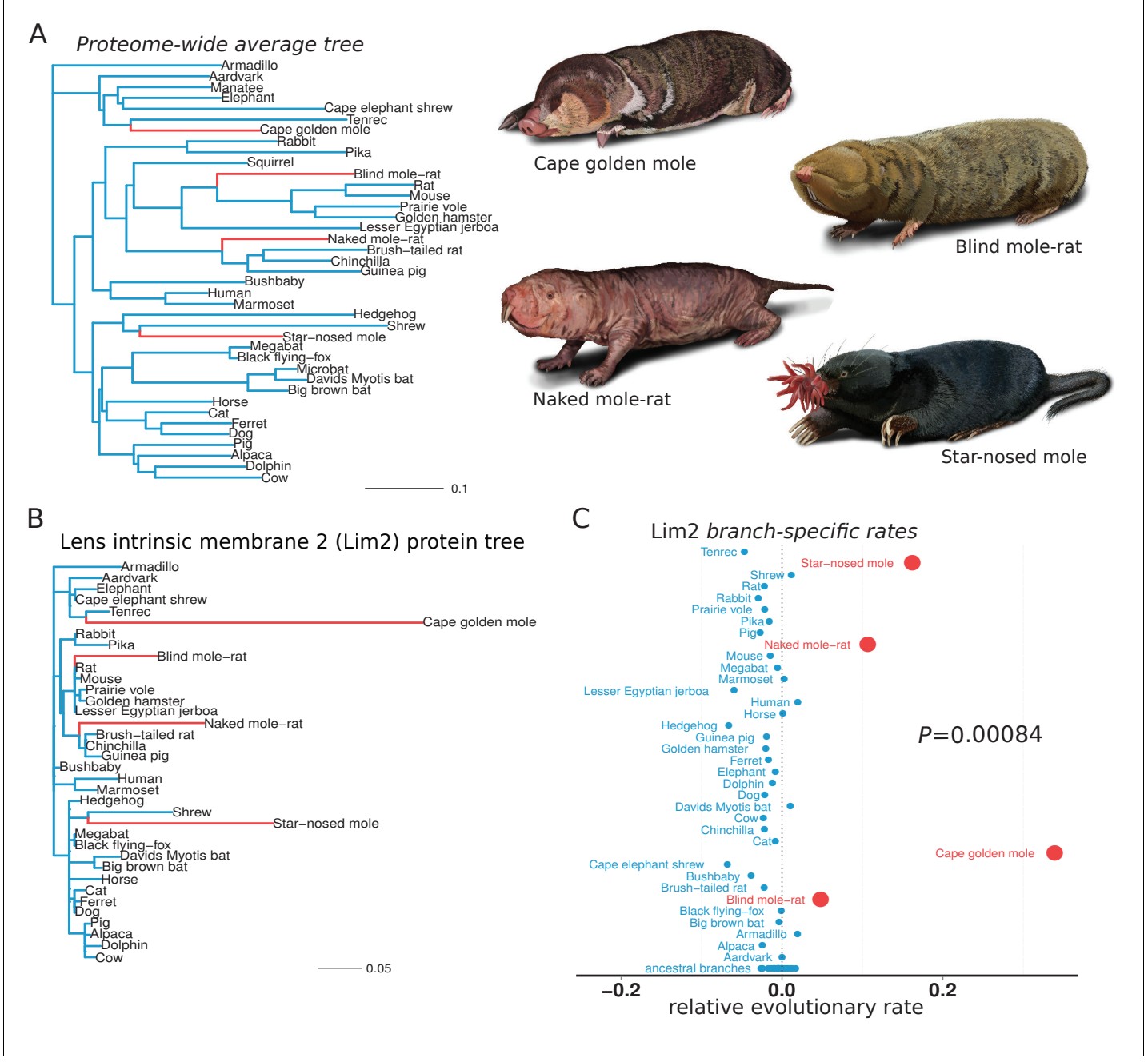

**Figure 1.** Lens intrinsic membrane protein 2 (LIM2) evolutionary rates across species. (**A**) Mammalian transitions to a subterranean environment occurred in four lineages shown in red. The branch lengths on the mammalian tree reflect the average evolutionary rate across 18,980 protein-coding genes. (**B**) LIM2 protein-coding sequence shows accelerated rates of evolution on subterranean branches compared to those on the proteome-wide average tree. (**C**) Relative evolutionary rates of LIM2 showed the strongest acceleration on the subterranean branches amongst all of the genes studied. Illustrations by Michelle Leveille (Artifact Graphics).

DOI: https://doi.org/10.7554/eLife.25884.003

annotated to the Gene Ontology (GO) term 'visual perception', because a large fraction of genes that have this annotation are ranked very highly in the list of mole-accelerated genes (*Figure 3A* 'subterranean'). Furthermore, if we were to employ mole-acceleration as a sole predictor of visual function, a search would correctly identify many known visual perception genes with high accuracy, even when searching the entire genome (*Figure 3B*). This strong enrichment allows us to pose

**Table 1.** Top 30 of 55 subterranean-accelerated genes.

| Gene | P-value | Tissues | Description |
|---|---|---|---|
| LIM2* | 0.00084 | Lens | Lens intrinsic membrane protein 2 |
| CRYBB3* | 0.00087 | Lens | Lens-specific crystallin, beta B3 |
| R0M1* | 0.00096 | Retina | Retinal outer segment membrane protein 1 |
| CRYBA1* | 0.00098 | Lens | Lens-specific crystallin, beta Al |
| CRYGC* | 0.00119 | Lens | Lens-specific crystallin, gamma C |
| CRYBB2* | 0.00128 | Lens | Lens-specific crystallin, beta B2 |
| GPR89B | 0.00130 | Ubiquitous | G-protein-coupled receptor 89B, pH mediator in Golgi |
| GNAT1* | 0.00133 | Retina | Rod cell-specific G-protein, subunit alpha |
| GPRS9A | 0.00134 | Ubiquitous | G-protein-coupled receptor 89A, pH mediator in Golgi |
| NRL* | 0.00138 | Retina | Neural retina leucine zipper responsible for expression of rhodopsin |
| CRYGS* | 0.00146 | Lens | Lens-specific crystallin, gamma S |
| GRM6* | 0.00150 | Retina | Metabotropic glutamate receptor 6, required for normal vision |
| GBX2 | 0.00165 | Embryo | Gastrulation brain homeobox 2, developmental transcription factor |
| LGSN* | 0.00171 | Lens | Lengsin, lens protein with glutamine synthetase domain |
| CRYBB1* | 0.00183 | Lens | Lens-specific crystallin, beta Bl |
| KLHDC3 | 0.00186 | Ubiquitous | Kelch-domain-containing 3, high expression in brain |
| KRT81# | 0.00186 | Hair and nails | Keratin 81, primarily in hair cortex |
| WDFY1 | 0.00192 | Ubiquitous | WD repeat and FYVE-domain-containing 1, endosomal protein |
| KRT9# | 0.00195 | Skin | Keratin 9, specific to palms of hands and soles of feet |
| POMP# | 0.00199 | Ubiquitous | Proteasome maturation protein, associated with rare skin disorder |
| RRH* | 0.00201 | Retina | Retinal pigment epithelium-derived rhodopsin homolog |
| DPCD* | 0.00201 | Ciliated cells | Deleted in primary ciliary dyskinesia; maintenance of ciliated cells |
| RAD54L | 0.00217 | Ubiquitous | RAD54-like: DNA double-strand break repair |
| TATDN1 | 0.00235 | Ubiquitous | TatD DNase-domain-containing 1 |
| ITLN2 | 0.00244 | Small intestine | Intelectin 2, may play a role in defense against pathogens |
| STX3* | 0.00245 | Ubiquitous | Syntaxin 3, associated with congenital cataracts and intellectual disability |
| SKJV2L* | 0.00254 | Ubiquitous | DEAD box protein, yeast SKI2 homolog, implicated in macular degeneration |
| DPY19L1 | 0.00254 | Ubiquitous | dpy-19-like 1 (*Caenorhabditis elegans*), probable C-mannosyltransferase |
| TFPT | 0.00266 | Ubiquitous | TCF3 (E2A) fusion partner (in childhood leukemia) |
| RSI* | 0.00275 | Retina | Retinoschisin 1, extracellular protein involved in organization of retina |

*related to vision.

#related to skin and hair.

Refer to **Supplementary file 1** for a full list of the subterranean-accelerated genes.

DOI: https://doi.org/10.7554/eLife.25884.006

specific hypotheses in subsequent sections about which tissues and genetic pathways were altered during the regressive evolution of the eye.

We performed a control analysis to demonstrate that these functional enrichments are unique to subterranean species. We chose four aboveground species (Control species) for which there is no reason to expect phenotypic convergence and whose branch lengths are similar to the moles – pika, guinea pig, squirrel and cow. Whereas mole-accelerated genes were enriched in 15 GO categories at a FDR of 15%,control-accelerated genes had no enriched categories at the same FDR (**Supplementary file 2**). Furthermore, these control species showed no enrichment of visual perception genes specifically (**Figure 3**).

There were also 1306 mole-decelerated genes that evolved at significantly slower rates in subterranean species than in other mammals (**Supplementary file 3**). Although mole-decelerated genes are individually significant, only one GO category showed significant functional enrichment – GO

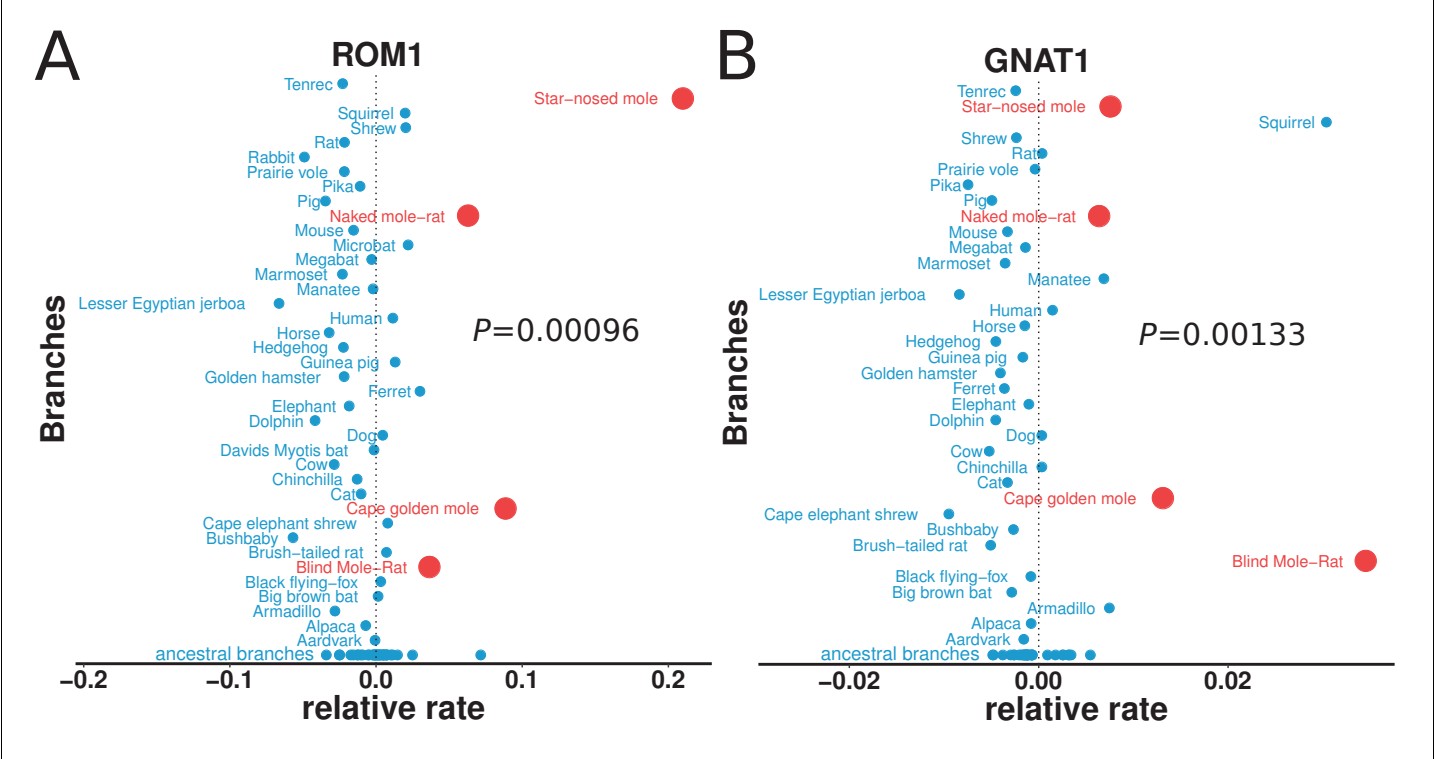

**Figure 2.** Relative evolutionary rates of two retinal proteins across species. Relative evolutionary rates of two retinal proteins, (A) Retinal outer segment membrane protein 1 (ROM1) and (B) Rod cell-specific G protein, subunit alpha (GNAT1), show strong acceleration in the subterranean mammals (marked in red).

DOI: https://doi.org/10.7554/eLife.25884.004

Biological Process: Nucleic acid binding transcription factor activity – at an FDR of 15% (*Supplementary file 4*). A similar control analysis showed 626 genes as being significantly decelerated at an FDR of 15%, and these control-decelerated genes were enriched in 24 GO categories. Therefore, despite there being vastly more mole-decelerated genes than mole-accelerated genes, mole-decelerated genes as a group do not show strong functional enrichment. This result stands in stark contrast to the strong enrichment seen in the mole-accelerated genes.

## Most mole-accelerated genes are under relaxed constraint

Accelerated rates could have resulted from adaptive evolution or, alternatively, from relaxation of constraint. We distinguished between these scenarios using codon-based evolutionary models to

**Table 2.** Representative enriched functions in mole-accelerated genes.

| Functional annotation | Fold enrichment | p-value | FDR q-value |
|---|---|---|---|
| Visual perception | 23.16 | 6.84E-16 | 1.02E-11 |
| Sensory perception of light stimulus | 22.69 | 9.12E-L6 | 6.82E-12 |
| Sensory perception | 8.47 | 5.83E-10 | 2.91E-06 |
| Neurological system process | 5.39 | 1.75E-07 | 6.53E-O4 |
| Detection of light stimulus | 29.57 | 7.04E-07 | 2.10E-03 |
| Detection of light stimulus involved in sensory perception | 56.35 | 1.92E-05 | 4.77E 02 |
| Detection of light stimulus involved in visual perception | 56.35 | 1.92E-05 | 4.09E-02 |
| Detection of external stimulus | 14.38 | 2 49E-05 | 4.66E-02 |

DOI: https://doi.org/10.7554/eLife.25884.007

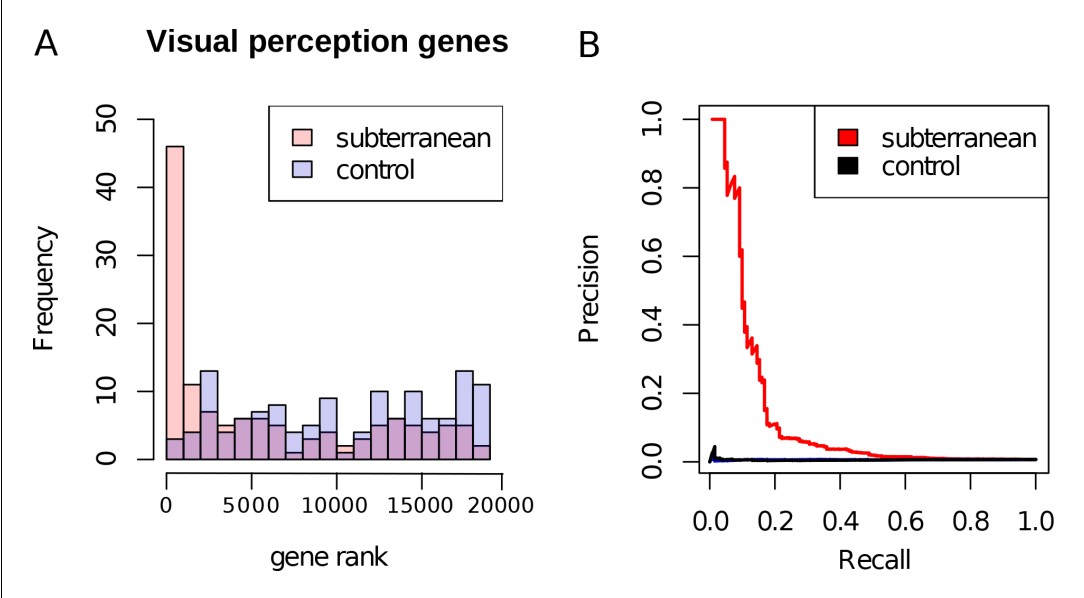

**Figure 3.** Enrichment of visual perception genes. (A) Histogram of the rankings of 189 visual perception genes based on their mole-acceleration. We see a clear enrichment of the genes with low rank numbers, reflecting the strong signal of mole-acceleration in visual perception genes. As a control, we use four non-subterranean species, and as expected, genes involved in vision do not show convergent rate acceleration. (B) Mole-acceleration can equivalently serve as a predictor for function in visual perception. The plot shows the Precision-Recall values at varying p-value thresholds reflecting the fraction of visual perception genes significant at a particular threshold (Precision) and the fraction of visual perception genes retrieved at the same threshold (Recall). We see that mole-acceleration specifically identifies visual perception genes with high precision when compared to acceleration in two sets of four non-subterranean control species.

DOI: https://doi.org/10.7554/eLife.25884.005

detect signatures of adaptive evolution. We tested whether the nonsynonymous to synonymous rate ratio ($d_N/d_S$) was significantly greater than 1 – the expectation for positive selection – for any portion of the gene specifically on the subterranean species branches, and also more generally across the entire mammalian phylogeny (*Yang, 2007*). Of the top 55 mole-accelerated genes, only one gene rejected a neutral model not allowing $d_N/d_S$ ratios exceeding 1 in favor of a model allowing positive selection ($d_N/d_S > 1$) on subterranean branches (*Supplementary file 5*). This gene is involved in connective tissue and hair structure (KRTAP17-1).

The other accelerated genes did not show evidence of adaptive evolution and thus are probably under relaxed constraint. Almost all accelerated genes rejected a model requiring them to have identical constraints in all mammals (model M1) in favor of a model that allowed subterranean-specific relaxation of constraint (model BS1) (*Supplementary file 5*). Some of these genes seem to have lost all functional constraint because they show genetic lesions such as stop codons and frameshifts in some subterranean species (*Supplementary file 6*). This evidence of relaxed constraint is consistent with the expectation that some vision-related genes have been undergoing regressive evolution.

## Skin-related genes were accelerated possibly in response to the demands of tunneling

The fossorial lifestyle of subterranean species has selected for traits related to digging and locomotion underground (*Nevo, 1979*). Perhaps because of this selective pressure, many of the top mole-accelerated genes encode proteins that are structural components of skin, hair and epithelial connective tissues. The reasons for their acceleration are the result of relaxation of constraint on their coding sequence. Genes encoding keratin proteins 9, 12, and 81 (KRT9, KRT12, KRT81) were studied using codon models, and the results indicated that they experienced relaxed constraint in subterranean species but not positive selection for amino acid diversification (*Supplementary file 5*). They

contain early stop codons in multiple subterranean species, which is consistent with complete loss of constraint (*Supplementary file 6*).

The convergent acceleration and pseudogenization of KRT9 is particularly interesting in relation to burrowing (*Figure 4*). In mice, KRT9 expression is confined to footpads, and $Krt9^{-/-}$ null mutants develop footpad calluses due to hyperproliferation of skin (*Fu et al., 2014*). In humans, KRT9 is expressed solely on the palms of hands and the soles of feet, and mutations lead to a skin disorder characterized by hyperkeratosis (thickening) of the surfaces of palms and soles – epidermolytic palmoplantar keratoderma (*Hennies et al., 1995*). By extension, the loss of KRT9 in subterranean species may also have led to hyperproliferation of footpads, which could carry benefits for tunneling. For example, the star-nosed mole digs with its forepaws, and naked mole-rats collect and remove dirt with their feet (*Jarvis and Sale, 2010*; *Hamilton, 1931*). Such abrasive tasks could place high demands on the footpad surfaces. In addition, mole-acceleration of the *POMP* gene could similarly have resulted from demands on footpads. A human mutation in *POMP* is associated with KLICK syndrome, a skin disorder also characterized by hyperproliferation and thickening of palms and footpads (*Dahlqvist et al., 2010*).

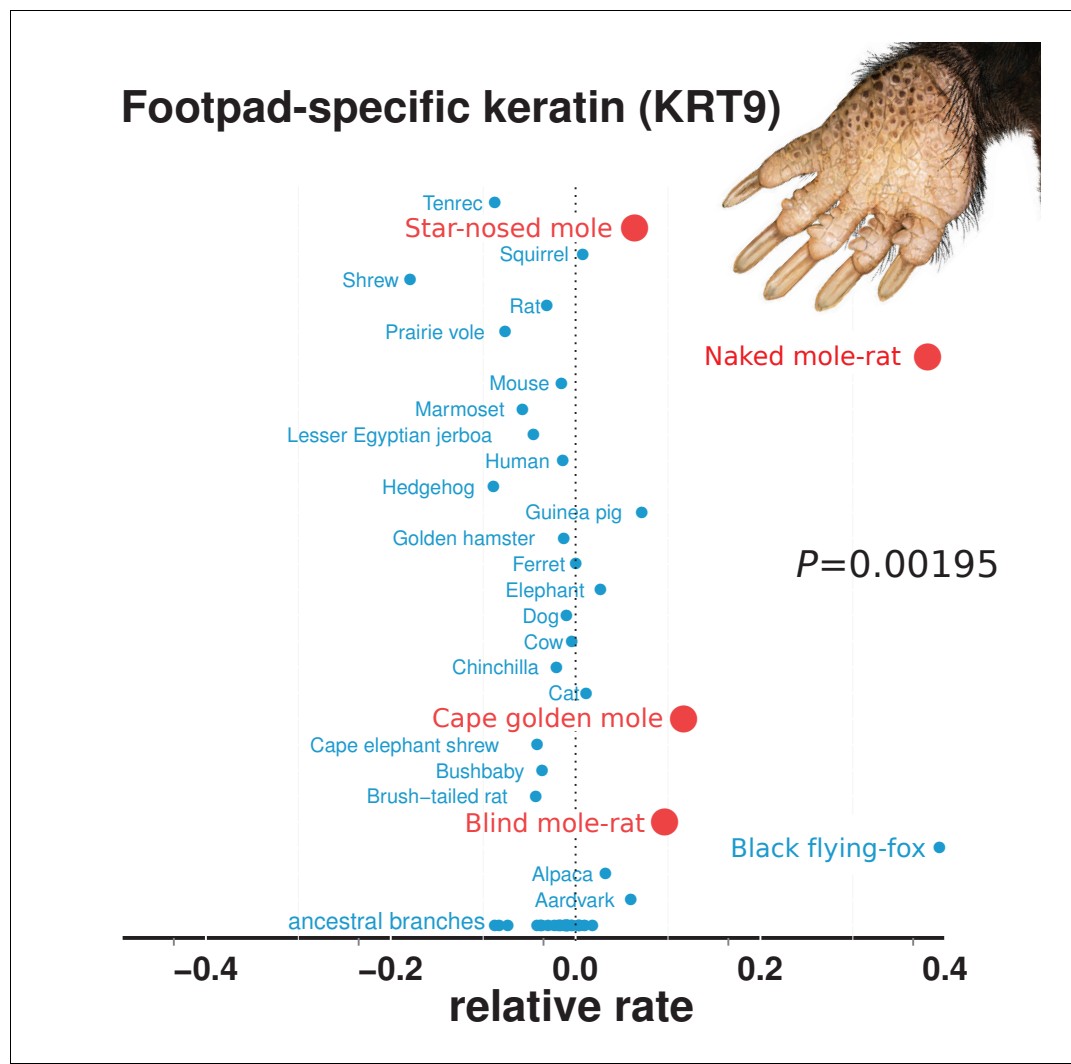

**Figure 4.** Relative rates of footpad-specific keratin 9 (KRT9). KRT9 shows strong acceleration on the subterranean branches. The image shown is the footpad of the star-nosed mole, showing characteristic hyperkeratosis. Keratin 9 mutations also lead to hyperkeratosis in mouse models and humans. Illustrations by Michelle Leveille (Artifact Graphics).
DOI: https://doi.org/10.7554/eLife.25884.008

In addition, we discovered skin- and hair-related genes showing evidence of positive selection rather than of loss of function in subterranean species. Although these genes were not significantly mole-accelerated at a FDR of 15%, they potentially reflect functional changes in response to subterranean adaptation. One such gene, *COL4A4*, a gene encoding a subunit of Type IV collagen, was strongly accelerated, did not contain genetic lesions, and showed evidence of positive selection in subterranean species (*Supplementary file 1,5,6*). Type IV collagen is the major structural component of the basal lamina in many tissues, including skin epithelium, and is composed of six subunits, three of which (COL4A4, COL4A5 and COL4A3) were notably mole-accelerated. On average, the six subunits were more accelerated than 71% of all other genes, which is a significant difference (p=0.0342, Mann-Whitney U test). Whereas Type IV Collagen seems to have responded to the subterranean environment, other major components of the basal lamina, the laminin proteins (e.g., LAMA1), were not notably accelerated.

## Regressive evolution is limited to the lens, retina, and eye-specific developmental genes

In order to compare how specific eye tissues have evolved in subterranean species, we first compiled tissue-specific gene sets using expression data from 91 mouse tissues (*Su et al., 2004*). We identified tissue-specific genes for cornea, iris, lens and retina by selecting those genes with significant differential expression in the tissue of interest but not in other tissues. Using literature, we also compiled a set of 71 important eye developmental genes (*Supplementary file 7*). We first asked whether there is a relationship between the degree of tissue-specificity and the degree of mole-acceleration measured as the difference in $d_N/d_S$ between subterranean and aboveground species (*Figure 5A*). We found a clear positive correlation between eye tissue-specificity and mole-acceleration, which is consistent with a greater relaxation of constraint on genes with few or no roles outside the eye. Next, we asked which genes with eye-tissue-specific expression showed acceleration and found that genes that are specifically expressed in the cornea (a protective tissue of the outer eye) and the iris were not accelerated in subterranean species when compared to a set of randomly chosen genes (background) (*Figure 5B and C*). By contrast, many lens- and retina-specific genes are

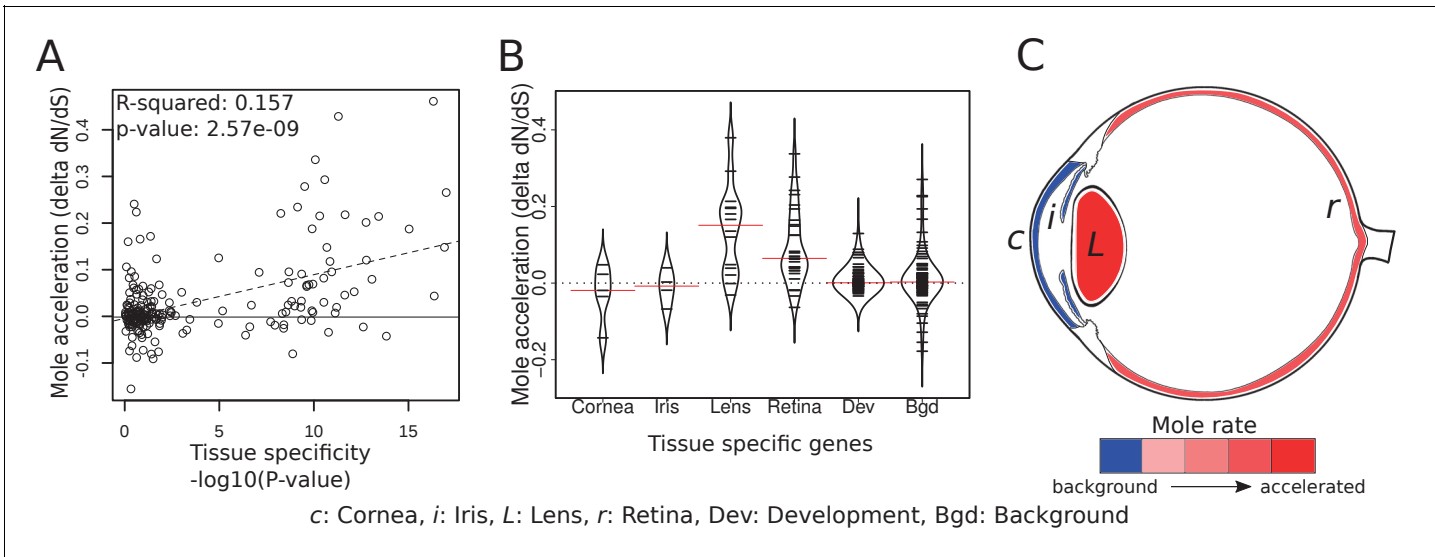

*c*: Cornea, *i*: Iris, *L*: Lens, *r*: Retina, Dev: Development, Bgd: Background

**Figure 5.** Tissue-specific retinal and lens genes are highly accelerated in subterranean species. (A) Ocular genes that are more tissue-specific exhibit stronger acceleration in subterranean 'mole' species. The y-axis represents the change in the rate of evolution on branches shifting to a subterranean environment. (B) Panels of tissue-specific genes were tested for their relative accelerations in the subterranean mammals. One hundred randomly chosen 'background' genes were not faster or slower on average, and provide an estimate of the variance expected for random genes. Retina- and lens-specific genes show many cases of acceleration in the subterranean environment, and their distributions are significantly elevated when compared to background (p=$1.4\times10^{-5}$ and $3.2 \times 10^{-4}$, respectively). (C) Representation of average mole-acceleration for genes specifically expressed in four different tissues of the eye.

DOI: https://doi.org/10.7554/eLife.25884.009

accelerated. On average, lens genes are more accelerated than 84% of background genes, and retina genes are more accelerated than 82% (p=9.07×10$^{-6}$ and p=6.10×10$^{-10}$ for lens and retina, respectively, Mann-Whitney U test). The contrast between the front and the interior of the eye suggests that the sensory functions of the inner eye, such as phototransduction and the visual cycle, are under relaxed constraint, whereas the protective function of the cornea is not. Indeed, two of these subterranean species have eyes that are open to the environment, such that the cornea may continue to serve as a barrier to pathogens and debris.

Eye developmental genes as a whole were not accelerated compared to background, which may reflect the fact that most of them, such as Sonic Hedgehog (*Shh*), are important in the development of non-eye tissues. However, five eye-specific developmental genes were notably present at the top of the accelerated list (*Vax2*, *N*rl, *Foxe3*, *Crx* and *Aldh3A1*), whereas no eye-specific genes were found lower in the list (*Supplementary file 7*). This is consistent with the positive relationship between eye-specificity and relaxation of constraint (*Figure 5A*).

## Eye-specific enhancers of *PAX6* show convergent acceleration in subterranean mammals

Although we observe specific instances in which eye developmental genes show accelerated rates in subterranean mammals, there is no significant global trend. This is understandable given that a majority of these developmental transcription factors have important roles in the development of non-eye-related tissues. For example, *Pax6* is important in the development of pancreas and brain in addition to the eye (*Kleinjan et al., 2006*; *Kammandel et al., 1999*; *Xu et al., 1999*). Hence the protein-coding sequences of the transcription factors encoded by these genes experience selective pressure against deleterious mutations. However, regulatory regions controlling the expression of these developmental genes in the eye might be under relaxed constraint in subterranean mammals, given the relaxation of the need to maintain the functionality of visual pathways. We hypothesize that these eye-specific *cis*-regulatory elements (CREs) would thus show accelerated rates of evolution in the subterranean mammals.

We tested this hypothesis by applying our evolutionary-based method toward identifying eye-specific regulatory elements controlling the expression of the developmental transcription factor PAX6. We chose the PAX6 system because extensive effort has gone into characterizing the spatio-temporal regulation of its expression (*Kleinjan et al., 2006*; *Kammandel et al., 1999*; *Xu et al., 1999*; *Kleinjan et al., 2001*; *Dimanlig et al., 2001*; *Griffin et al., 2002*), and there exists comprehensive annotation ofCRE that control the expression of PAX6 in various tissues including the eye. On the basis of existing literature on the transcriptional regulation of *Pax6* expression, we identified a 500-kb window containing *Pax6* and its neighboring gene *Elp4* as our genomic window of interest (*Kleinjan et al., 2006*). Experiments involving transgenic mice revealed various tissue-specific enhancers in a 200-kb region within this genomic window to be important for *Pax6* expression. We subsequently identified 150 highly conserved non-coding elements in this genomic window and estimated their evolutionary rates on each mammalian branch. We then calculated the relative rates of the branches using the same projection operator method as was employed for the protein-coding gene trees. We then employed the Mann-Whitney U hypothesis-testing framework to identify non-coding elements that have evolved at an accelerated rate specifically on the subterranean branches (Materials and methods).

The results of our analyses show that the three regions showing the strongest signals of convergent acceleration in the subterranean mammals extensively overlap the regions previously annotated to be enhancers important for regulation in eye-specific tissues (*Figure 6A, B*). (i) 'cre149' is a 558-basepair (bp) region containing the 530-bp region annotated as the 'alpha, intron 4 retinal' enhancer (*Kammandel et al., 1999*). (ii) 'cre21' is a 552-bp region located within the fragment containing HS2 and HS3 of the Distal Regulatory Region, a retina-specific enhancer of PAX6 (*Kleinjan et al., 2001*). (iii) 'cre86' is a 429-bp region containing the 341-bp long 'ectodermal enhancer', which has been shown to be important in driving the expression of PAX6 in the developing lens (*Dimanlig et al., 2001*). Regions overlapping an enhancer element shown to be regulating PAX6 expression in lens, hindbrain and diencephalon (the 'EI' enhancer element) do not show significant rate acceleration in the moles (*Kleinjan et al., 2001*). This is in concordance with our expectation that only eye-specific elements show convergent acceleration, and hence the regions overlapping the EI enhancer do not show acceleration given their importance for PAX6 expression in non-eye tissues. Similarly, a 120-bp

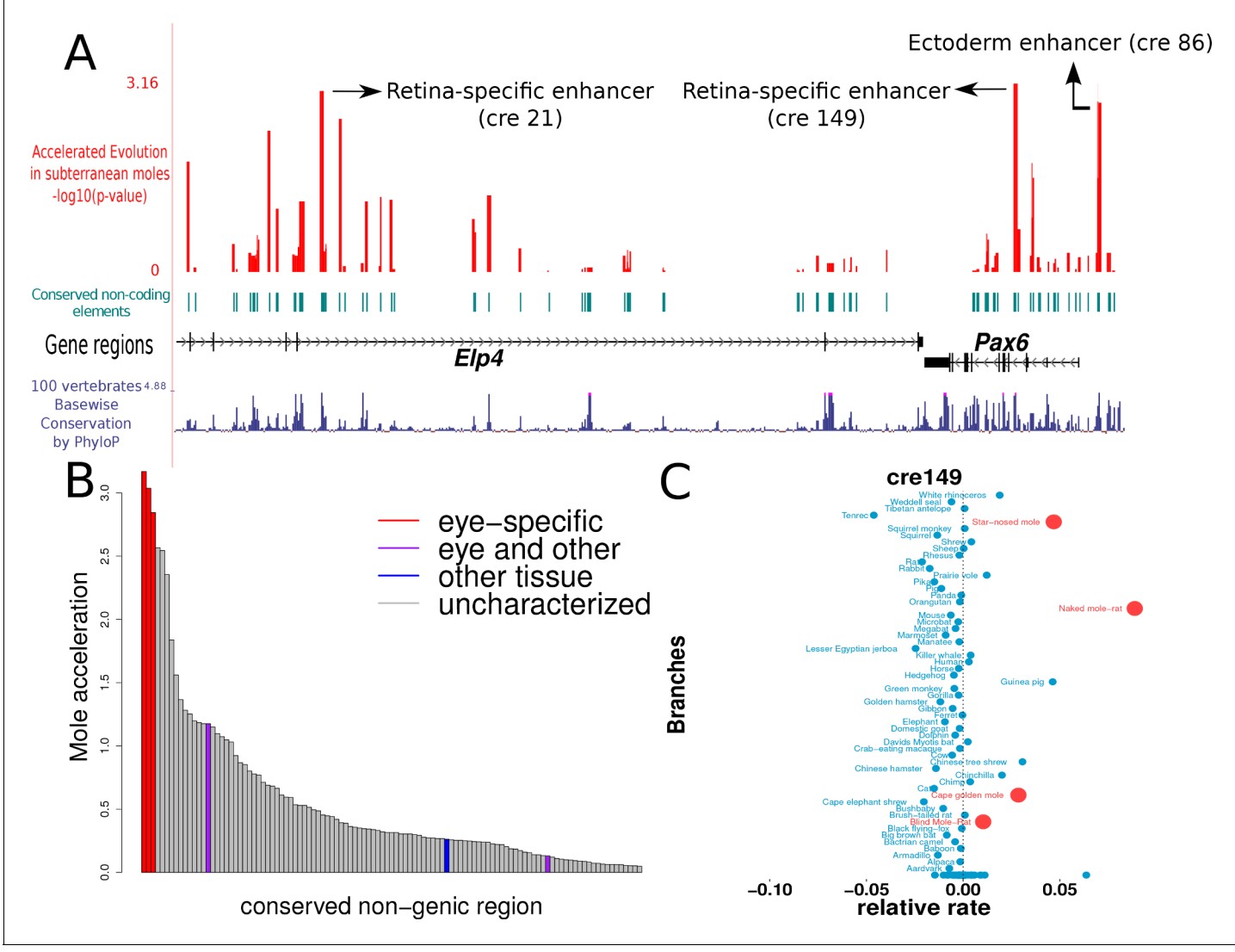

**Figure 6.** Mole-acceleration of eye-specific enhancers in the *Pax6* gene region. (A) Genomic region spanning *Pax6* and its neighbor *Elp4*. The exons and introns of the two genes are represented by black blocks and lines respectively, whereas the conserved non-coding regions analyzed are represented in light blue. The conservation signal as given by the 100 vertebrates Basewise Conservation is shown in dark blue. The mole-acceleration scores for these regions are represented in red. The three most accelerated non-coding regions identified in this analysis are consistent with the eye-specific enhancers regulating *Pax6* expression in the eye. (B) The mole-acceleration scores for the three eye-specific enhancers of *Pax6* are the highest among 150 regions analyzed, including enhancers of other tissues and uncharacterized non-coding regions. (C) The relative rates in each species for the most accelerated region 'cre149'.

DOI: https://doi.org/10.7554/eLife.25884.010

region overlapping the pancreas enhancer also does not show significant rate acceleration in the moles, as expected (*Kleinjan et al., 2006*; *Xu et al., 1999*). In addition to the eye-specific enhancer elements, we observe other regions showing comparable rate acceleration in the moles that are not yet characterized (*Supplementary file 8*). These regions are candidate CREs for PAX6 expression in the eye. This preliminary study of the *Pax6* transcriptional regulatory module serves to confirm our hypothesis that eye-specific regulatory elements are under relaxed constraint and thus show accelerated rates of evolution in the subterranean mammals.

## Mole-accelerated non-coding elements are strongly enriched near transcription factor genes driving eye development

Expanding from our analysis of *Pax6*, we performed a large-scale scan for convergently accelerated non-coding elements near transcription factor genes in the mammalian genome. We compiled two sets of transcription factor genes – one comprising 20 genes known to be important in eye development (the Eye set), such as *Pax6*, *Pax2*, *Otx2*, and another set consisting of an equal number of tissue-specific transcription factor genes that are expressed in other tissues and with no evidence of expression in eye (the Other set), which includes *Hoxa9*, *Pax8* and *Sox13* (*Supplementary file 9*). We identified 200 conserved non-coding elements near each gene in both sets, totaling to 8000 elements split equally between the two gene sets (see Materials and methods). We subsequently applied our method and calculated the mole-acceleration of each element. This large-scale scan revealed a total of 17 elements as convergently accelerated at an FDR of 10% (*Figure 7A*, *Supplementary file 10*). Fourteen of the 17 elements are found near to genes belonging to the Eye set, reflecting a significant enrichment of mole-accelerated elements near transcription factor genes driving eye development (Hypergeometric test, p-value=0.001). We subsequently checked the genomic locations of these mole-accelerated elements to ensure that they are not clustered at the same locus for instance. These 17 elements are found close to 14 unique genes, with 11 unique genes belonging to the Eye set, and three genes belonging to the Other set, further showcasing the strong enrichment of unique eye developmental transcription factor genes close to mole-accelerated elements (Hypergeometric test p-value = 0.0016).

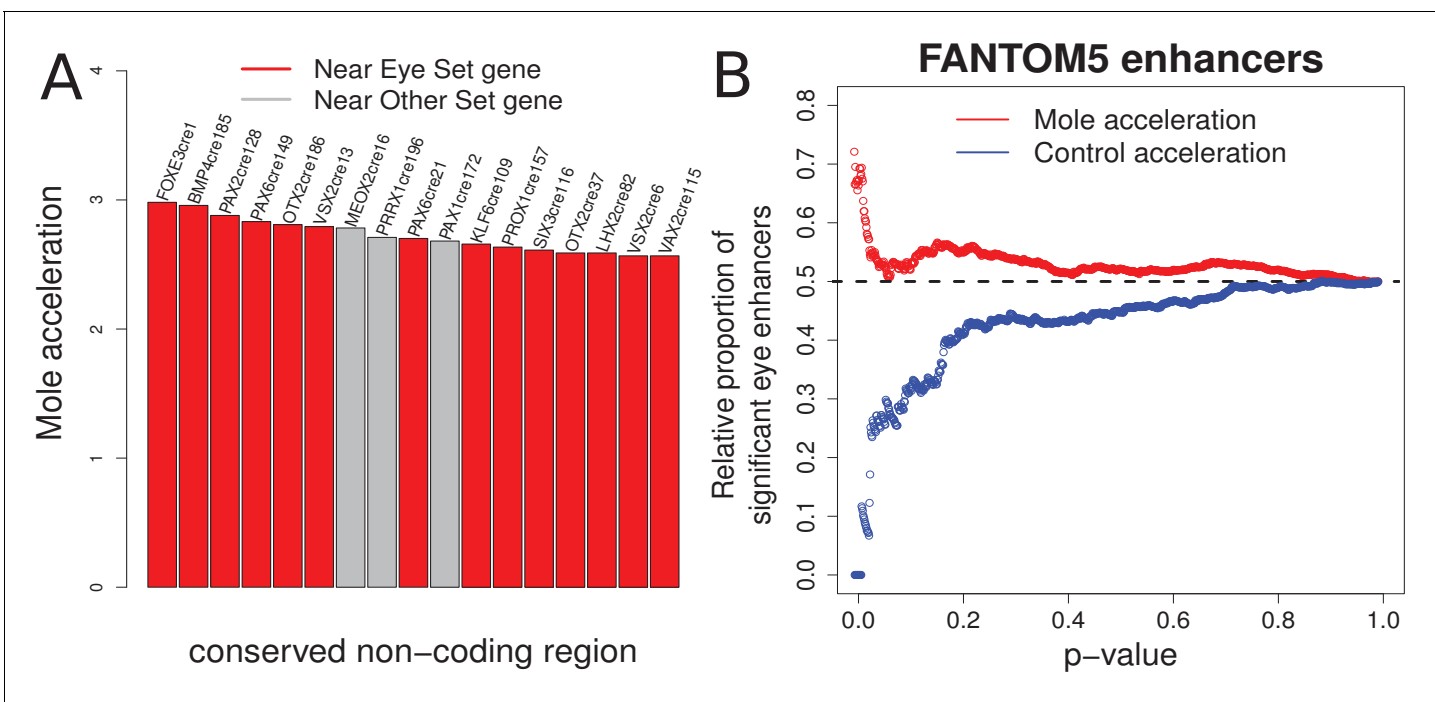

**Figure 7.** Evidence of mole-acceleration in candidate eye-specific enhancers. (**A**) Enrichment of mole-accelerated elements near eye developmental transcription factor genes. The bar plot shows the 17 mole-accelerated conserved non-coding elements identified. Fourteen of the 17 elements are present near transcription factor genes in the Eye set, denoted in red. (**B**) FANTOM5 Eye enhancers show strong mole-acceleration. The plot shows the relative proportion of FANTOM5 eye enhancers identified among all enhancers significant at the corresponding *p*-value threshold. We see a strong enrichment of eye enhancers identified at low mole-acceleration *p*-values (red points) whereas no such enrichment is observed using control-species-acceleration p-values (blue points).

DOI: https://doi.org/10.7554/eLife.25884.011

## FANTOM5 eye enhancers show strong convergent acceleration in subterranean mammals

The FANTOM5 consortium has identified putative enhancer sites in the human and mouse genomes based on bidirectional enhancer transcription across tissues as well as at multiple developmental time points (*Andersson et al., 2014*). These putative enhancer sites include genomic regions that are transcribed in the eyeball of mouse embryos at four developmental time points. On the basis of this resource, we compiled two sets of FANTOM5 enhancer sites (see Materials and methods) – a set consisting of 900 genomic regions with non-zero expression in the eyeball across four developmental time points ('Eye' enhancers), and another set consisting of 6000 regions with zero expression across the same samples ('Other' enhancers). We subsequently calculated the convergent rate acceleration of these genomic elements in the four subterranean mammals and compared the acceleration observed for the 'Eye' enhancers to that of the 'Other' enhancers. Our analysis revealed a strong enrichment of FANTOM5 'Eye' enhancers showing convergent rate acceleration in the four subterranean species when compared to the four control species (*Figure 7B*). We observe 62 FANTOM5 enhancers in total that showed significant mole acceleration at an FDR of 15% (*Supplementary file 11*). Fifteen of these correspond to the FANTOM5 Eye enhancers set, reflecting a significant enrichment of detected FANTOM5 eye enhancers using mole-acceleration (Hypergeometric test p-value=0.006).

## Some aboveground species exhibit gene acceleration indicative of their altered visual capacities

To understand differences in the visual capabilities of mammals systematically, we studied the overall relative rates of evolution of visual genes across all mammals. Our gene set of interest (189 genes in total) was comprised of all genes with 'Visual perception' GO term annotation, excluding developmental transcription factors. For each species, we then calculated the mean relative rate across all of the genes (*Figure 8*). We observed the four subterranean mammals to be among the accelerated species (with mean >0), as was our expectation. However, we also observed aboveground species with overall rate accelerations comparable to those of the moles, such as the armadillo, the thirteen-lined ground squirrel, the big brown bat, David's myotis bat and a shrew. Notably, all of these mammals show varying types of visual regression: the armadillo has poor vision characterized by a lack of cone cells in the retina (*McDonough and Loughry, 2013*), and shrews also have poor vision and diminutive eyes, which in some species are hidden in fur (*Nowak, 1999*). The nocturnal big brown bat and David's myotis bat possess reduced eyes and rely on echolocation for navigation (*Koay et al., 1998*). The thirteen-lined ground squirrel displays a rare visual trait: the central region of its retina is dominated by cone photoreceptors in contrast to the retinas of most mammals (*Kim et al., 2016*). These scenarios could have important implications because the ground squirrel is used as a model for vision research (*Li et al., 2010*; *Chen and Li, 2012*).

## Discussion

The independent transitions of four mammals to a subterranean environment has been accompanied by convergent phenotypic changes that have arisen as a result of adaptation to new environmental stresses in the underground ecotope (*Nevo, 1979*; *Leys et al., 2003*; *Lacey et al., 2000*; *Albert et al., 2007*; *Wilkinson, 2012*). Here, we report a genome-wide effort encompassing both coding and regulatory regions to identify the changes in genotype that have accompanied this phenotypic adaptation by studying changes in their evolutionary rates. Our study reveals that genes showing convergent acceleration in subterranean species are highly enriched for function in visual pathways. The decreased selective pressure on visual pathways in the dim-light subterranean environment leads to a relaxation of constraint on genetic elements involved in various eye-related phenotypes, including eye morphology, photoreception and visual transduction. In addition to accelerated change in genes in visual pathways, we observe an accelerated rate of evolution of many genes involved in skin-related phenotypes in the subterranean mammals. Whereas we see accelerated change in visual genes primarily as a result of relaxation of constraint, we see that some skin-related genes also show accelerated change due to positive selection, perhaps as a result of selection of traits contributing to a fossorial lifestyle. Aside from these two phenotypes, we

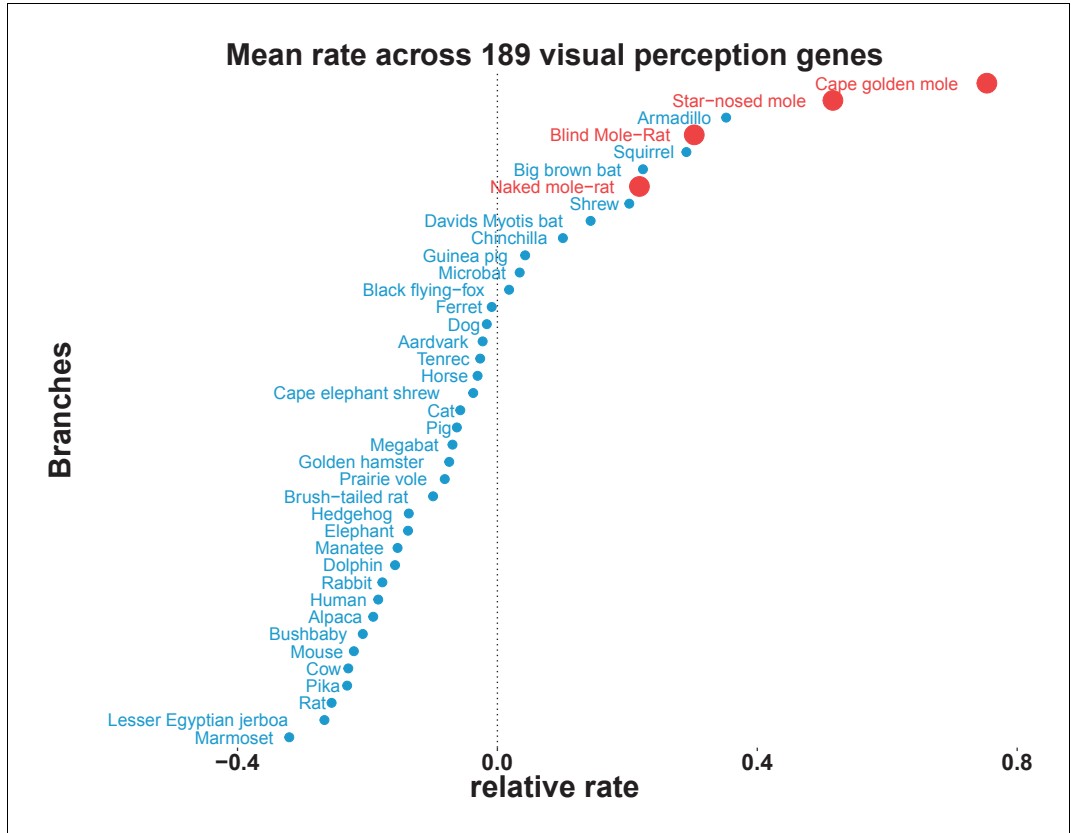

**Figure 8.** Some aboveground species show accelerated rates of evolutionary change in visual perception genes. On the basis of the relative evolutionary rates across all species for 189 genes with the GO term annotation 'visual perception', we calculated the species-wise mean relative rate across of the genes. Our previous observations of mole-acceleration in visual perception genes are recapitulated here – the four subterranean mammals are among the species that show an accelerated rate across these genes. Interestingly, we find other non-subterranean species showing acceleration comparable to the subterranean mammals, indicating adaptations in visual systems.

DOI: https://doi.org/10.7554/eLife.25884.012

do not observe a comparably strong enrichment for genes involved in the other environmental challenges associated with a subterranean lifestyle, such as hypoxia, hypercapnia and high infectivity. It is possible that the subterranean mammals may show species-specific adaptations to these stresses, whereas our analysis from a convergent evolutionary perspective reflects changes common to all the species.

Closer examination of the accelerated genes that are enriched for vision-related pathways reveals that accelerated genes tend to be lens- or retina-specific. On the other hand, genes encoding specifically for the outer ocular structure, the cornea, do not show significant acceleration, indicating the preservation of developmental programs that are important for ocular architecture. In two of the four moles with non-subcutaneous eyes, the cornea can come into direct contact with the external environment, perhaps necessitating the proper development of the structure in the highly infective subterranean niche. Lens- and retina-specific genes that are involved with the processes of photoreception and phototransduction would be under greater relaxed constraint given the dim-light environment, accruing damaging mutations at a much higher rate. Our analyses also reveal that genes that are associated with congenital eye diseases are accelerated in the four subterranean mammals. For the lens, which is largely made up of crystallins, we find many crystallin genes (*Crybb3*, *Cryba1*, *Crybb1*, *Crygc*, *Crygs*, etc.) in our accelerated set of genes contributing to various forms of cataracts (*Graw, 2009*). Similarly, we find multiple genes involved in ciliopathies to be accelerated, including 'deleted in primary ciliary dyskinesia' (*Dpcd*), *Iqcb1* (a component of primary cilia), and ciliary neurotrophic factor (*Cntf*). Further inspection of the accelerated list of genes could potentially reveal new candidate genes that are important for congenital eye diseases.

Genes that are involved in the embryonic development of eyes do not show significant global acceleration, potentially due to their pleiotropic nature: these developmental transcription factors tend to have important regulatory roles in non-eye-related pathways that are not under relaxed constraint. We successfully tested our hypothesis that eye-specific regulatory elements of such genes are under relaxed constraint in the moles, using a novel variant of our approach that calculates convergent rate acceleration at the non-coding level. Although the strong rate acceleration in the three eye-specific enhancers of PAX6 suggests relaxation of constraint in the subterranean mammals, in the absence of functional tests we cannot be sure that the eye-specific activity is truly lost. Furthermore, we found an enrichment of such convergently accelerated non-coding regions preferentially near eye developmental transcription factor genes, identifying potential enhancer elements driving the expression of these genes specifically in the eye. As a large-scale validation approach, we show that rate acceleration in subterranean mammals strongly overlaps regions identified as eye enhancers by the FANTOM5 consortium. These proof-of-principle analyses serve to illustrate the power of convergent-evolution-based tools for the identification of eye-specific regulatory elements. Despite the apparent rapid rate of enhancer evolution across mammals, our methods and those of colleagues showcase the utility of applying evolution-based approaches to conserved non-coding regions in identifying regulatory elements underlying important developmental functions (*Marcovitz et al., 2016*; *Villar et al., 2015*). These methods present a unique opportunity to perform genome-wide scans for eye- and other tissue-specific regulatory elements, and potentially serve as complementary approaches to genome-wide assays in the identification of active enhancer elements in the genome. As more genomes are sequenced, we expect these methods to become more powerful in revealing gene regulatory changes underlying convergent phenotypes.

Overall, our results suggest that genes and non-coding regions that are involved in vision pathways are accumulating deleterious mutations by neutral processes, given the relaxation of constraint on these pathways in the subterranean environment. However, this does not preclude the possibility that the initial inactivating mutations in these pathways were adaptive in nature. The initial shutdown of eye development may have been caused by positively selected changes, followed by continued regression of structural and physiological eye genes through neutral processes. Indeed, there is evidence of such a progression of events during eye regression in blind cavefish (*Jeffery, 2005*). Adaptive forces for reduced eyes may have been driven by the energetic costs of maintaining functioning eyes and the risk of pathogen entry through the eye (*Moran et al., 2015*). We note that our rate-based analysis detects signatures of sequence divergence that are based on what is observed at the end of these processes and does not shed light on the nature of the initial inactivating changes. In addition, our methods detect convergent changes in the rates of evolution of genes and hence are not designed to detect species-specific changes that might contribute to the subterranean adaptation.

Our results showcasing acceleration in the rates of convergent evolution of visual genes strongly supports previous reports of visual regression in the subterranean habitat. *Emerling and Springer (2014)* studied the regression of retinal genes in three of these four subterranean species and showed that a decrease in the amount of light entering the retina is associated with higher incidence of inactivating mutations in retinal genes. They found a significantly higher number of retinal pseudogenes in the moles compared to that in closely related subaerial species, an observation concordant with our results based on rate acceleration. Genome sequencing efforts for naked mole-rat and blind mole-rat also showed a strong enrichment of pseudogenes in visual pathways that are associated with the degradation of vision in these species (*Cooper et al., 1993*; *Kim et al., 2011*). A genome-wide study by *Prudent et al. (2016)* detected significant genomic differences in genes involved in vision-related pathways such as eye development and perception of light in two of these four subterranean mammals, namely cape golden mole and blind mole-rat. Using our rates-based framework, we performed a rigorous investigation of convergently evolving genes in a large set of four subterranean species, and elucidated the tissue-specificity and underlying reasons for their convergent rate changes. In a first-of-its-kind demonstration at the non-coding level, we applied our methods successfully to detect eye-specific enhancers showing accelerated evolution in subterranean mammals.

Visual regression is not limited to these four mole species, and mammals display specific types of regression and other general differences in visual capabilities. Our analysis of visual gene rates across other species revealed interesting patterns and trends, wherein some aboveground species

with poor or remodeled visual systems showed mean rate acceleration comparable to subterranean mammals (*Figure 8*). This provides an opportunity to further probe specific differences in the development and function of visual systems in terms of the specific pathways that are relaxed or under constraint across species. In addition, integrating these other species into our rate-based framework can help in fine-tuning the predictive power of the evolutionary-based approaches. Deliberate selection of foreground branches based on specific combinations among these vision-impaired mammals might greatly improve the power of the methods in detecting convergent changes, especially at the non-coding level. In this regard, the availability of rich and diverse phenotypic annotations across mammals further lays the ground for the development of evolutionary-based approaches in functional and phenotypic annotation of non-coding regions (*Marcovitz et al., 2016*; *O'Leary et al., 2011*; *O'Leary et al., 2013*).

## Materials and methods

### Adding *Nannospalax galili* orthologs to alignment

Given the absence of *Nannospalax galili* (blind mole-rat or BMR) in the 100-species alignments made available by the UCSC genome browser, we employed a custom approach to add the correct BMR orthologous sequence based on its closest relative on the mammalian species phylogeny, mouse. Using the publicly available BMR gene models (ftp://ftp.ncbi.nlm.nih.gov/genomes/all/GCF_000622305.1_S.galili_v1.0/), we first perform ed pairwise reciprocal nucleotide blast of all BMR gene cDNA sequences and the corresponding cDNA sequences of all genes in the mouse mm9 genome. For every mm9 gene sequence, we subsequently identify the correct BMR ortholog using the InParanoid program as follows: the program clusters pairs of sequences from the two queried genomes into groups of orthologs, and the BMR sequence forming the main ortholog pair (pairs with mutually best hit) in every group was identified as the correct ortholog (*Remm et al., 2001*). We then performed a profile alignment using the openly available Muscle program to add the identified BMR ortholog to the gene's multi-species alignment (*Edgar, 2004*). For all analyses involving non-coding regions, we utilized a simpler approach to identify the BMR orthologous region. For each non-coding region of interest, we performed blastn with the mm9 orthologous sequence as the query against the BMR assembled genome, with the default Expect (E) value of 10 (*NCBI Resource Coordinators, 2016*). The resulting best scoring blastn hit in the BMR genome, if any, was added to the non-coding region's multi-species alignment (obtained from the UCSC genome browser) using the profile alignment utility of the Muscle program (*Edgar, 2004*).

### Calculating gene correlations with the subterranean environment

Using the 100-way 100 vertebrate species amino acid alignments from the multiz alignment available at the UCSC genome browser (*Blanchette et al., 2004*; *Harris, 2007*), those alignments with a minimum of 10 species that are also present in at least two subterranean species were selected for the study. We pruned each alignment to include only the 39 species of interest represented in the proteome-wide average tree (*Figure 1A*), after adding the BMR ortholog of the corresponding gene sequence to this alignment as described in the previous section. For each resulting amino acid alignment, we estimated branch lengths using the 'aaml' program from the phylogenetic analysis using the maximum likelihood (PAML) package (*Yang, 2007*). Branch lengths were estimated under an empirical model of amino acid substitution rates with rate variability between sites modeled as a gamma distribution approximated with four discrete classes (for computational efficiency) and an additional class for invariable sites (aaml model 'Empirical + F') (*Whelan and Goldman, 2001*; *Yang, 1996*). Branch lengths were estimated on a published mammalian species tree topology (*Murphy et al., 2004*), modified to include *Nannospalax galili* whose position in the tree was inferred based on existing literature on its ancestry (*Fang et al., 2014*). For the analyses involving conserved non-coding elements, we identified the elements of interest based on the human phastCons track generated from the 100-way vertebrate multiz alignment, eliminating any region of overlap with the human mRNAs track. For each such element, we obtained an alignment of orthologous regions across our species of interest, pruning from the UCSC 100-way multiz alignment. The previous section further details the procedure employed for adding the BMR orthologous region. We subsequently estimated the branch lengths using the baseml program of the PAML package under

the general reversible process (REV) model for nucleotide substitution rates, with rate variability between sites modeled as a gamma distribution approximated with four discrete classes and an additional class for invariable sites (*Blanchette et al., 2004*; *Rodríguez et al., 1990*).

Subsequent to reconstructing the maximum likelihood trees using PAML, we filter out the trees of genetic elements that have zero branch lengths in at least 80% of the species present in the tree. Raw branch lengths in trees retained after this filtering step were transformed into relative rates using a projection operator method (*Sato et al., 2005*). These branch-specific relative rates were then used to perform a Mann-Whitney U test and correlation analysis over the binary variable of 'subterranean' or 'aboveground' (i.e., not subterranean) branches (*Figure 1A*). Subterranean branches are those leading to the star-nosed mole (*Condylura cristata*), cape golden mole (*Chrysochloris asiatica*), naked mole-rat (*Heterocephalus glaber*) and blind mole-rat (*Nannospalax galili*).

## Datasets of 'Eye' vs 'Other' conserved non-coding elements and FANTOM5 enhancers

### A. Conserved non-coding elements near transcription factor genes

We identified 20 developmental transcription factors that have important developmental roles in the formation of eye tissues ('Eye' set) based on a literature search. The detailed functional roles of these transcription factors and the specific eye tissues where they are relevant are provided in *Supplementary file 9*. The second part of the dataset comprises 20 transcription factors identified as belonging to the 'Other' set. These transcription factors have no known role in eye development and their tissue-specific functions were identified from a census of human transcription factors [*Supplementary file 9*] (*Vaquerizas et al., 2009*). Subsequently, we used the UCSC phastConsElements100way track to identify conserved non-coding elements near each transcription factor gene. For each gene, we identify 200 elements expanding the search window from the center of the gene along either direction. We limit the number of elements to 200 in order to avoid any biases arising out of the total number of elements studied near any particular gene. This leads to a total of 8000 elements split equally between the 'Eye' and the 'Other' set.

### B. FANTOM5 enhancers

We downloaded a dataset of 44,460 putative enhancers identified by the FANTOM5 consortium, including their mm9 coordinates and expression quantification across 1190 tissue samples (*Andersson et al., 2014*). These include 1217 enhancers with non-zero expression level in the eyeball of mice ('Eye') across four developmental time-points and 13,100 enhancers with zero expression in eyeball ('Other'). We obtained the corresponding hg19 coordinates of these enhancers using the UCSC liftOver utility (*Kent et al., 2002*). Based on this, we were able to map correctly 995 enhancers belonging to the 'Eye' set and 7695 enhancers belonging to the 'Other' set. From among these enhancers, we were able to confidently calculate the evolutionary rate correlation with the subterranean environment for 946 FANTOM5 'Eye' enhancers and 6,331 FANTOM5 'Other' enhancers, after filtering out enhancers that were either poorly conserved across our species set or whose trees were dominated by branches of length zero.

## Functional enrichment analysis

We performed functional enrichment analysis using the GOrilla tool by searching for enriched GO terms in the foreground set of genes compared to the full background set of genes tested for mole convergence (*Eden et al., 2009*). In addition to this, functional information for subterranean-associated genes was mined from the Uniprot and RefSeq databases, and from literature cited directly (*UniProt Consortium, 2007*; *Pruitt et al., 2007*). Enrichment analysis was performed using the hypergeometric test with the background set of genes restricted to genes that were tested for mole convergence and that had at least one annotation in the corresponding annotation file. Correction for multiple testing was performed using false discovery rate q-values (*Storey, 2002*).

## Multiple hypothesis testing correction

False Discovery Rate analysis was performed on probabilities resulting from the Mann-Whitney U test. We employed an empirical permutation-based FDR calculation, often the standard approach in genome-wide analysis. We generated 10,000 null datasets, obtaining each dataset by randomly

permuting the species labels of the relative rates. This process is equivalent to calculating the Mann-Whitney U test and correlation analysis over four random branches vs the rest instead of over the binary variable of 'subterranean' vs 'aboveground' branches. The subsequent permuted datasets were used to estimate the FDR q-values for each p-value in our subterranean correlation analysis. Genes showing a correlation greater than or equal to 0.5 in the Mann-Whitney U test and significant at a FDR of 15% were considered mole-accelerated genes, and the p-value reflecting the strength of this acceleration is referred to as mole-acceleration. Similarly, genes showing a correlation less than 0.5 and significant at a FDR of 15% are considered mole-decelerated genes.

### Tissue-specific gene analysis

In order to determine how specific eye tissues have evolved across subterranean species, we first identified tissue-specific gene sets using microarray expression data from 91 mouse tissues (*Su et al., 2004*). We isolated tissue-specific genes for cornea, iris, lens and retina (including retinal pigmented epithelium). These sets were defined as those with significant differential expression only in the tissue of interest compared to all other tissues at an alpha of 0.05 (T-test).

### Phylogenetic models of selective pressure

The subterranean-accelerated genes were subjected to phylogenetic models of codon evolution to test for significant evidence of relaxation of constraint or positive selection over the subterranean mammal branches. Using PAML, we ran *codeml* using five different models: the branch-site neutral model (BS Neutral), the branch-site selection model (BS Alt Mod), the sites neutral model (M1), the positive selection model (M8) and its null model (M8A) (*Yang, 2007*). To assess the significance of relaxation of constraint on subterranean mammal branches, we performed likelihood ratio tests (LRT) between BS Neutral and its nested null model M1. LRTs between BS Alt Mod and its null BS Neutral were used to infer positive selection on subterranean mammal branches. Probabilities were assigned for each of these two LRTs using the chi-square distribution with 1 degree of freedom. Mammal-wide positive selection was inferred using the M8 vs M8A models and their respective LRT, using 1 degree of freedom chi square distribution to assess LRT significance. For calculating the correlation between mole-acceleration and degree of tissue-specificity of genes, we estimated the mole-acceleration of each gene as follows: using a branch-site selection model (BS Alt Mod) we estimated two different values of $\omega$ ($d_N/d_S$) – one for the four subterranean branches and one for the rest of the branches on the tree. Mole-acceleration was calculated as the difference in the two $\omega$ values that were estimated.

## Acknowledgements

We thank John Wible of the Carnegie Museum for discussion of visual phenotypes. This study was supported by a Charles E Kaufman New Investigator Award from The Pittsburgh Foundation to NLC, National Institutes of Health (NIH) grants (R01HG009299 and U54HG008540) to MC and NLC, NIH grant R03MH109009 to MC, a National Eye Institute/National Institutes of Health Core Grant (P30 EY008098), unrestricted grants from Research to Prevent Blindness, Inc., the Jack Buncher Foundation, and the Eye and Ear Foundation of Pittsburgh to BKC and KKN, and a Research Experience for Undergraduates grant funded by the National Science Foundation and the Department of Defense (NSF- DBI 1263020).

## Additional information

### Funding

| Funder | Grant reference number | Author |
| --- | --- | --- |
| National Eye Institute | Core Grant P30 EY008098 | Bharesh K Chauhan<br>Ken K Nischal |
| Research to Prevent Blindness | | Bharesh K Chauhan<br>Ken K Nischal |
| Eye and Ear Foundation of Pittsburgh | | Bharesh K Chauhan<br>Ken K Nischal |

| | | |
|---|---|---|
| Jack Buncher Foundation | | Bharesh K Chauhan<br>Ken K Nischal |
| National Institutes of Health | Core Grant P30 EY008098 | Bharesh K Chauhan<br>Ken K Nischal |
| National Science Foundation and Department of Defense | Research Experience for Undergraduates NSF-DBI 1263020 | Joseph D Robinson |
| National Institutes of Health | U54HG008540 | Maria Chikina<br>Nathan L Clark |
| National Institutes of Health | R03MH109009 | Maria Chikina |
| National Institutes of Health | R01HG009299 | Maria Chikina<br>Nathan L Clark |
| Pittsburgh Foundation | Charles E. Kaufman New Investigator Award | Nathan L Clark |

The funders had no role in study design, data collection and interpretation, or the decision to submit the work for publication.

### Author contributions

Raghavendran Partha, Zelia Ferreira, Conceptualization, Data curation, Software, Formal analysis, Visualization, Writing—original draft, Writing—review and editing; Bharesh K Chauhan, Data curation, Formal analysis, Funding acquisition, Writing—original draft, Writing—review and editing; Joseph D Robinson, Conceptualization, Data curation, Software, Formal analysis, Funding acquisition; Kira Lathrop, Visualization; Ken K Nischal, Funding acquisition, Writing—review and editing; Maria Chikina, Conceptualization, Software, Formal analysis, Funding acquisition, Writing—review and editing; Nathan L Clark, Conceptualization, Software, Formal analysis, Funding acquisition, Visualization, Writing—original draft, Writing—review and editing

### Author ORCIDs

Raghavendran Partha http://orcid.org/0000-0002-7900-4375
Bharesh K Chauhan https://orcid.org/0000-0002-6429-9190
Zelia Ferreira http://orcid.org/0000-0002-7619-7466
Nathan L Clark http://orcid.org/0000-0003-0006-8374

### Decision letter and Author response

Decision letter https://doi.org/10.7554/eLife.25884.031
Author response https://doi.org/10.7554/eLife.25884.032

## Additional files

### Supplementary files

• Supplementary file 1. Subterranean-accelerated genes The genome-wide ranking of genes whose rates are positively associated with the subterranean branches was identified on the basis of the Mann-Whitney U test p-values (column 2) that were significant at a FDR of 15% under a permutation-based FDR correction procedure (column 3). Other columns list brief gene descriptions (4), known biological function (5), tissue specificity if any (6), known disease links (7), the predominant evolutionary mode on subterranean branches as decided with codon models (8, see Materials and methods), and the subterranean species containing genetic lesions in the corresponding protein-coding sequences (9).
DOI: https://doi.org/10.7554/eLife.25884.013

• Supplementary file 2 Functional enrichment in subterranean-accelerated genes The subterranean-accelerated genes were strongly associated with specific Gene Ontology terms representing functional categories. This table presents the GO term identifier (column 1), description (column 2), the p-value representing the significance of the degree of enrichment of genes in that category among mole-accelerated genes relative to the entire set (3), the multiple test-corrected p-value (4), the

degree of enrichment in the accelerated set relative to the entire set (5), the number of genes in that category in the entire analyzed gene set (6), the number of genes in the accelerated set (7), the number of genes in that category in the accelerated gene set (8), and the corresponding gene names (9).

DOI: https://doi.org/10.7554/eLife.25884.014

• Supplementary file 3. Subterranean-decelerated genes The genome-wide ranking of genes whose rates are negatively associated with the subterranean branches was identified based on the Mann-Whitney U test p-values (column 2) that are significant at a FDR of 15% under a permutation-based FDR correction procedure (column 3). Column 4 lists brief gene descriptions.

DOI: https://doi.org/10.7554/eLife.25884.015

• Supplementary file 4. Functional enrichment in subterranean-decelerated genes The specific Gene Ontology terms representing functional categories with which subterranean-decelerated genes were strongly associated. This table presents the GO term identifier (column 1), description (column 2), the p-value representing the significance of the degree of enrichment of genes in that category among mole-accelerated genes relative to the entire set (3), the multiple test-corrected p-value (4), the degree of enrichment in the accelerated set relative to the entire set (5), the number of genes in that category in the entire analyzed gene set (6), the number of genes in the accelerated set (7), the number of genes in that category in the accelerated gene set (8), and the corresponding gene names (9).

DOI: https://doi.org/10.7554/eLife.25884.016

• Supplementary file 5. Results of *codeml* tests for positive selection in subterranean-accelerated genes This table presents the results for the mole-accelerated genes from five codon models and the likelihood ratio tests between them. Columns B – T present the branch-site models and tests with their log likelihood values, the likelihood ratio tests between them (see Materials and methods), and some of the estimated parameters for background (conserved) and foreground (subterranean) branches (*Zhang et al., 2005*). Columns V – AC present the results of the pan-mammalian tests for positive selection using the sites models (M8A and M8) (*Swanson et al., 2003*; *Yang, 2007*).

DOI: https://doi.org/10.7554/eLife.25884.017

• Supplementary file 6. Pseudogene study This table lists the pseudogene status of mole-accelerated genes' orthologs in the set of species utilized for the study.

DOI: https://doi.org/10.7554/eLife.25884.018

• Supplementary file 7. Eye developmental gene evolutionary rates. This table presents the evolutionary rates of the 71 eye developmental genes. For each gene (column 1), we performed a branch-site selection model analysis to estimate two different values of dN/dS – one for the four subterranean branches on the tree (3), and one for the rest of the branches (2, see Materials and methods). The mole-acceleration is calculated as the difference between these two values (4). Column 5 indicates whether the gene was identified as a subterranean-accelerated gene using the RER methods, and column 6 lists the refSeq summary.

DOI: https://doi.org/10.7554/eLife.25884.019

• Supplementary file 8. Relative rate acceleration in subterranean mammals for 150 conserved non-genic regions near *Pax6* This table presents the 150 non-coding regions scanned near *Pax6* and their relative rate acceleration in subterranean mammals. The first three columns represent the hg19 coordinates of the region. Column 4 represents the mole-acceleration calculated as the negative logarithm to the base 10 of the Mann-Whitney U test p-value. Column 5 provides additional information about the overlap of the region with known tissue-specific enhancers of *Pax6*.

DOI: https://doi.org/10.7554/eLife.25884.020

• Supplementary file 9. Transcription factor genes utilized for genome-wide scan for convergently accelerated non-coding elements This table presents the two sets of transcription factor genes used to scan for convergently accelerated non-coding elements. TF genes in the Eye set, were compiled based on a literature survey of their eye developmental role (column 2). Column 3 lists the refSeq summary. Similarly, the tissue-specificity of TF genes belonging to the 'Other' set is provided in column 2 of the second table.

DOI: https://doi.org/10.7554/eLife.25884.021

- Supplementary file 10. Relative rate acceleration in subterranean mammals for conserved non-coding regions near developmental transcription factor genes This table presents the 17 subterranean-accelerated non-coding regions scanned near the 40 TF genes listed in *Supplementary file 9*, significant at a FDR of 10%. Columns 1–3 list their hg19 coordinates. Column 4 presents the mole-acceleration calculated as the Mann-Whitney U test p-value. Column 5 lists the permutation-based FDR q-value.

DOI: https://doi.org/10.7554/eLife.25884.022

- Supplementary file 11. Relative rate acceleration in subterranean mammals for putative FANTOM5 enhancer regions This table presents the 62 subterranean-accelerated putative FANTOM5 enhancers significant at a FDR of 15%. Fantom5 element names containing the word 'eye' correspond to FANTOM5 enhancers identified as eye enhancers in this analysis. Columns 2 and 3 list each region's mm9 and hg19 coordinates respectively. Column 4 represents the mole-acceleration calculated as the Mann-Whitney U test p-value. Column 4 lists the permutation-based FDR q-value.

DOI: https://doi.org/10.7554/eLife.25884.023

- Transparent reporting form

DOI: https://doi.org/10.7554/eLife.25884.024

## Major datasets

The following previously published datasets were used:

| Author(s) | Year | Dataset title | Dataset URL | Database, license, and accessibility information |
|---|---|---|---|---|
| Kent WJ, Sugnet CW, Furey TS, Roskin KM, Pringle TH, Zahler AM, Haussler D | 2002 | The human genome browser at UCSC | http://hgdownload.soe.ucsc.edu/goldenPath/hg19/multiz100way/ | Freely available for use at the dataset URL |
| Fang X, Nevo E, Han L, Levanon EY, Zhao J, Avivi A, Larkin D, Jiang X, Feranchuk S, Zhu Y, Fishman A, Feng Y, Sher N, Xiong Z, Hankeln T, Huang Z, Gorbunova V, Zhang L, Zhao W, Wildman DE, Xiong Y, Gudkov A, Zheng Q, Rechavi G, Liu S, Bazak L, Chen J, Knisbacher BA, Lu Y, Shams I, Gajda K, Farré M, Kim J, Lewin HA, Ma J, Band M, Bicker A, Kranz A, Mattheus T, Schmidt H, Seluanov A, Azpurua J, McGowen MR, Ben Jacob E, Li K, Peng S, Zhu X, Liao X, Li S, Krogh A, Zhou X, Brodsky L, Wang J | 2014 | Genome-wide adaptive complexes to underground stresses in blind mole rats Spalax | https://www.ncbi.nlm.nih.gov/assembly/GCF_000622305.1 | Publicly available at NCBI Assembly (accession no: GCA_000622305.1) |
| Su AI, Wiltshire T, Batalov S, Lapp H, Ching KA, Block D, Zhang J, Soden R, Hayakawa M, Kreiman G, Cooke MP, Walker JR, Hogenesch JB | 2004 | A gene atlas of the mouse and human protein-encoding transcriptomes | https://www.ncbi.nlm.nih.gov/geo/query/acc.cgi?acc=GSE1133 | Publicly available at the NCBI Gene Expression Omnibus (accession no: GSE1133) |

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
