## [Decision Letter]

Thank you for submitting your article "Subterranean mammals show convergent regression in ocular genes and enhancers, and adaptation in tunneling-related genes" for consideration by *eLife*. Your article has been reviewed by three peer reviewers, and the evaluation has been overseen by a Reviewing Editor and Diethard Tautz as the Senior Editor. The following individuals involved in review of your submission have agreed to reveal their identity: Leslie M Turner (Reviewer #3).

The reviewers have discussed the reviews with one another and the Reviewing Editor has drafted this decision to help you prepare a revised submission.

Summary:

The authors employ a sequenced-based evolutionary rate approach to identify genomic regions under accelerated or decelerated evolution in four species of subterranean mammals, thereby associating convergent genetic changes in these species with phenotypic traits (with a focus on loss of vision). Their results show an enrichment of vision-related gene annotations among genes showing convergent accelerated evolution in subterranean mammals, most of which show signatures of relaxed constraint, rather than adaptation. By comparing accelerated genes to either genes showing eye-specific gene expression or genes involved in eye development, the authors further suggest that regression of eye function associates with relaxed constraint on genes expressed in the eye lens and retina. Lastly, they demonstrate the applicability of this approach to non-coding regions by analysing rates of evolution among subterranean mammals for non-coding regions in the Pax6 locus. Based on these results the authors propose that the regions they identify include novel candidate sequences for congenital eye disorders.

The reviewers agreed that your analysis of the genetic evolution occurring in four separate lineages of subterranean mammals has potential to be of wide interest, and so the reviewers and handling editor are requesting resubmission. However, the two serious concerns outlined below must be exhaustively addressed, or a revised manuscript will be rejected.

Essential revisions for resubmission:

1) Presentation and novelty (see below for specifics).

All reviewers and editors agreed that your current manuscript needs serious editorial revision to address pervasive logic and presentational problems that results in masking of the story's novelty. Many problematic examples are listed below, compiled from all three reviewer's comments. A specific subpoint that was extensively discussed was whether your manuscript was sufficiently novel for publication in *eLife*, given the recent publication of a highly similar analysis by Prudent et al. of two of the same species in MBE. The additional power of using two more species must be far better argued, or a revision will not be adequate for *eLife* publication.

2) Genome-wide analysis of enhancers (see below for specifics).

The final computational analysis that dissected Pax6 enhancers must be extended to a genome-wide analysis of all enhancers to identify the complete set of regions where fossorial pressures have resulted in genetic changes. This analysis should likely be an entirely new figure that compellingly displays the genome-wide data in a way that is broadly understandable and persuasive in highlighting the novelty over Prudent et al. MBE.

The title is accurate but a bit awkward: authors please suggest your own revision.

Further detail on points above:

1) Presentation and novelty (specific details).

Note that the concerns listed below arose from all four reviews, so some are not in exact occurrence order. In addition, the Reviewing Editor had great difficulty understanding what gene sets were used, when, why, and how – throughout the manuscript. The entire revision must be heavily revised to didactically lay out each analysis more slowly and clearly to non-expert readers.

In Table 1 and Supplementary file 1, Tables S1 to S6, the authors continually shift between the top 1000, 500 and 200 accelerated or decelerated genes. For clarity, it would be better to decide on a threshold of statistical significance and state in the main text how many accelerated and decelerated genes are detected using this threshold. The authors could also homogenise the set of genes used in supplementary tables in Supplementary file 1, unless there are specific advantages to using different sets.

Related to the above, how does the RER method compare to the one used by Prudent et al.?

Figure 3 could additionally use as control a set of convergent genes in a different subset of species, such as the one reported by the authors for marine adaptations (Chikina et al., 2016).

The supplemental excel tables in Supplementary file 1 could be better annotated, for example by adding legends at the top. Table S1 in Supplementary file 1 seems to be duplicated in the excel file.

Why does Table S7 in Supplementary file 1 only have 71 genes, given that the authors mention in the main text their using 98 eye developmental genes? It would be helpful to mark significantly accelerated genes in this table.

Wording in the subsection “Regressive evolution is limited to the lens, retina, and eye-specific developmental genes” is somewhat misleading and should be revised. ("we asked which eye tissues showed acceleration", whereas I think the authors are actually testing whether genes with eye-specific expression are enriched among accelerated genes).

While the acceleration that the authors observe in Pax6 eye enhancers is consistent with relaxed constraint or "regression", in the absence of functional tests of these enhancers it is possible that the function of these enhancers is intact. In contrast, many of the vision genes that exhibit convergent sequence alteration have elevated dN/dS ratios or have stop codons and frame shift mutations that provide clear evidence of relaxed constraint. I don't feel that the authors need to perform functional tests of the Pax6 eye enhancers of subterranean species, but they should mention this caveat in the Discussion section.

In the Introduction, the authors cite Fang et al. as providing evidence for convergent evolution of circadian rhythm genes in blind mole-rat and naked mole-rat genomes. However, a corrigendum for this paper reported an error in the amino acid sequence alignment of the CLOCK genes. After correcting this error, there was no evidence of CLOCK protein sequence convergence between blink mole-rat and naked mole rat. Thus, it seems that Feng et al. may not provide support for the claim that circadian rhythm genes have cover gently evolved similar sequence changes. The authors should adjust their statements in the introduction accordingly. Fang et al., 2014. Nat Commun 5:1039-1052 for Corrigendum see http://www.nature.com/articles/ncomms9051

The authors repeatedly use the term "non-genic" to refer to non-coding sequences. The use of this term is somewhat ambiguous. For instance, the Pax6 gene has known (as well as putative) enhancer elements that are located within introns of the Elp4 gene. Thus, those elements lie within the Elp4 gene and could be considered "intra-genic" relative Elp4. The term "non-genic" should be replaced with the term "non-coding" throughout the manuscript. I also suggest that the authors replace "genic" with "coding," since the term "genic" can refer to the entire gene (not just coding exons, but also promoter regions, 5'UTR, 3'UTR, etc.).

Gene and protein nomenclature: There are some errors and inconstancies in gene and protein names. For instance, "Lim2" is used to refer to the "LIM2" protein. "Pax6" is used instead of "PAX6." Also, when referring to "krt-/-" mutants, the first letter should be capitalized and the gene name should be italicized.

Figure 3: The gene rank labeling in the histogram is truncated and needs to be extended to include the bars on the right end of the chart.

Figure 5: The abbreviation "Dev" and "Bgd" need to be defined in the figure legend.

Figure 6: The species that is used to diagram the gene region should be specified. If the species is mouse, then mouse gene nomenclature should be followed when labeling the diagram (first letter capitalized, name in italics).

Introduction includes long, highly detailed lists of adaptations – hard to follow and main points are lost. A figure or table summarising adaptations observed in subterranean taxa would be better.

Writing is wordy and colloquial in many places. e.g. "Even though it is possible to visualize this accelerated rate of change in the Lim2 tree, it is more rigorous to use a quantitative framework."

Results include a lot of interpretation, sometimes quite speculative. As a result, much of the Discussion is repetitive.

Overselling of enhancer results: I think the Pax6 enhancer results are a compelling example, suggesting regulatory elements may show similar convergent relaxed constraint to coding genes. However, title, Abstract, and Discussion suggest genome-wide analysis of enhancers was performed and found visual-specific enhancers accelerated ("successfully proved our hypothesis").

Adaptation in tunneling-related genes: title and in the subsection “Skin-related genes were accelerated possibly in response to the demands of tunneling”. “reasons for their acceleration are split equally between relaxation of constraint and positive selection.” No numbers are given for # positively selected and total # tested – I believe only one positive selection example is listed (COL4A4). How is 'skin-related' defined – which GO terms?

Significance of acceleration/deceleration.

What is p-value threshold considered 'significant'?

How was multiple testing accounted for?

P values for top 1000 accelerated and decelerated in Supplementary file 1, Table S1/S3 very different – 614 accelerated have P <0.05; max for 1000 decelerated is P = 0.018.

How does P value distribution compare to 2 control sets used for GO analysis? Or random sets of 4 species?

Rods vs. cone genes analysis (subsection “Each subterranean species exhibited selective gene acceleration consistent with its visual capacities”)

It is hard to evaluate these results because I can't find a legend explaining ++s and --s in Supplementary file 1, Table S8 (top) and there are several empty cells in the pseudogene section (bottom). Nevertheless, this section is overly descriptive and I don't think much can be concluded based on inconsistent patterns across 3 genes/category.

Some Methods are unclear:

Beginning of Materials and methods should clearly explain overall rationale for data included in RER. e.g., describe 100-spp alignment, criteria for inclusion in proteome average tree, minimum # of subterranean sequences required for analysis (vs. 10 overall).

Figure 3 – how was this analysis done? What counts as 'mole acceleration' and 'visual perception'?

Arbitrary/inconsistent gene list #s. Why top 30 genes in Table 1? Top 500 accelerated in GO analysis Table 2? Why 1000 top genes included in supplementary tables in Supplementary file 1? Why 200 included in PAML analysis?

Pax6 enhancer analysis – (subsection “Eye-specific enhancers of PAX6 show convergent acceleration in subterranean mammals”, second paragraph), if enhancers in 200 kb window, why 500 kb 'window of interest', is 200 kb in center? Specify locations.

Tissue-specific expression (subsection “Regressive evolution is limited to the lens, retina, and eye-specific developmental genes”). It is unclear what 'significant differential expression only in the tissue of interest' means – significantly higher expression only in one tissue compared to background?

Figure 1 – include scientific names.

Figure 1, Figure 2, Figure 4: what are unlabeled blue dots at bottom? Why are species ordered alphabetically by common name (it appears, not specified) instead of by phylogeny?

Figure 5 – r2? P value?

Figure 5 – not described in legend; shades hard to distinguish.

Figure 6 – indicate 200 kb and 500 kb windows for analysis.

Subsection “Most subterranean-accelerated genes are under relaxed constraint” – how many mole-accelerated genes show lesions in 1 or more subterranean species?

Subsection “Regressive evolution is limited to the lens, retina, and eye-specific developmental genes”, “clustered at the top of the accelerated list” – what does this mean? Overall list in Supplementary file 1, Table S1?

2) Extending Pax6 enhancer analysis (specific details)

The analysis on non-coding regions (Figure 6) must be expanded. If the point the authors want to make is that their approach is applicable to non-coding regions, this approach should be applied to all conserved non-coding elements rather than a single locus. An increased scope of this analysis would add novelty to the author's findings. Can one still detect an enrichment of eye-related gene annotations proximal to enhancers with accelerated evolution in subterranean mammals? How does it compare with that of coding regions?

The authors highlight their search for convergent patterns of sequence divergence as a means to identify genetic elements that may underlie changes in phenotype. If the authors analyzed each subterranean mammal on its own (looking at sequence divergence but not convergence between subterranean species), do they see strong enrichment of vision-related GO terms in each species? Do they also see other non-vision GO terms (e.g., hypoxia)? How different are the GO enrichments for accelerated proteins in individual subterranean species vs. the list of proteins that show convergent accelerations among subterranean species?

3) Other (non-required) points

As a strategy to inform the molecular basis of phenotypic traits, the author's approach is an alternative to direct experimental mapping across mammals (e.g. of gene expression; Ma et al., 2016). The authors hypothesize that genomic regions under accelerated evolution in subterranean mammals may inform candidate loci for congenital eye disorders, but no direct evidence is provided. The manuscript would tremendously benefit from substantiating this hypothesis with experimental data, for example by evaluating a few of the accelerated loci (genes or regulatory regions) in animal models of eye development, such as mouse or zebrafish. Can the authors show a requirement for some of their novel candidate elements in eye development or eye-specific expression? (Please see (Kvon et al., 2016) for a similar approach).

---

## [Author Response]

Essential revisions for resubmission:1) Presentation and novelty (see below for specifics).All reviewers and editors agreed that your current manuscript needs serious editorial revision to address pervasive logic and presentational problems that results in masking of the story's novelty. Many problematic examples are listed below, compiled from all three reviewer's comments. A specific subpoint that was extensively discussed was whether your manuscript was sufficiently novel for publication in eLife, given the recent publication of a highly similar analysis by Prudent et al. of two of the same species in MBE. The additional power of using two more species must be far better argued, or a revision will not be adequate for eLife publication.

We thank the reviewers for identifying areas where the manuscript could be improved upon and have addressed each point to the best of our ability. Regarding the question of novelty over Prudent et al., there are a number of reasons our manuscript presents a major advance over their paper:

1) Our method makes use of 4 species compared to the two used by Prudent et al. and hence can more precisely and confidently identify genetic elements convergently responding to the subterranean environment.

2) We perform a more detailed investigation including studying the reasons for the rate acceleration in these genes as well as where they were expressed across specific tissues of the eye. On the other hand, Prudent et al.’s presentation of mole evolution was a short description. Their presentation seems to have fulfilled its purpose of demonstrating their new methods but was not intended to reveal biological insights.

3) To highlight the contrast in the performance of our methods, we conducted a head-to-head comparison of the Prudent method to ours on a common dataset of 3 mole species. We found that ours performs better on both protein-coding and enhancer sequences. For full details see our second response to point 1 below. In the revised manuscript however, we chose to remain focused on the biological conclusions and did not include this comparison of methodologies. If the reviewers feel the comparison to Prudent et al. is crucial for publication, we are happy to add those results to the manuscript.

4) In a first-of-its-kind demonstration we show the applicability of our methods on non-coding regions, which we view as a more urgent challenge, and correctly identify experimentally validated eye-specific enhancers of Pax6. Prudent et al. did not report mole-acceleration for non-coding sequences.

5) Additionally, we now include two large-scale non-coding analyses covering 16,000 potential enhancer sites across the genome to illustrate the power of our method in identifying candidate vision-specific genetic elements. We also provide the chromosomal coordinates of the top-scoring regions in a Supplemental Table for the research community.

We believe that the combination of the aforementioned points clearly illustrates the advancement of our findings in comparison to those reported in the Prudent et al. publication in MBE.

2) Genome-wide analysis of enhancers (see below for specifics).The final computational analysis that dissected Pax6 enhancers must be extended to a genome-wide analysis of all enhancers to identify the complete set of regions where fossorial pressures have resulted in genetic changes. This analysis should likely be an entirely new figure that compellingly displays the genome-wide data in a way that is broadly understandable and persuasive in highlighting the novelty over Prudent et al. MBE.

We have now performed two large-scale scans for convergently mole-accelerated non-coding elements, totaling to 16,000 in number. These two scans give rise to the following findings – 1) Mole-accelerated genetic elements are strongly enriched near transcription factors with known function in eye development compared to tissue-specific transcription factors outside the eye; 2) Eye enhancers identified by the FANTOM5 consortium show significant mole-acceleration compared to other Fantom5 enhancers with zero expression in the eyeball of developing mouse embryos. These new analyses strongly reflect the power of our method in identifying putative vision-specific non-coding elements, illustrating the first successful attempt at large-scale prediction of vision-specific genetic elements using a convergent-evolution-based approach. For the gene regulation research community, we provide the coordinates of these newly described mole-accelerated non-coding reqions (Supplementary file 1, Table S10). In addition to this, we perform a direct comparative analysis that showed better performance of our RER method over the Prudent et al. method as part of our response to point 1.

The title is accurate but a bit awkward: authors please suggest your own revision.

We have clarified the title.

“Subterranean mammals show convergent regression in ocular genes and enhancers, along with adaptations to tunneling”

Further detail on points above:1) Presentation and novelty (specific details).Note that the concerns listed below arose from all four reviews, so some are not in exact occurrence order. In addition, the Reviewing Editor had great difficulty understanding what gene sets were used, when, why, and how – throughout the manuscript. The entire revision must be heavily revised to didactically lay out each analysis more slowly and clearly to non-expert readers.In Table 1 and Supplementary file 1, Tables S1 to S6, the authors continually shift between the top 1000, 500 and 200 accelerated or decelerated genes. For clarity, it would be better to decide on a threshold of statistical significance and state in the main text how many accelerated and decelerated genes are detected using this threshold. The authors could also homogenise the set of genes used in supplementary tables in Supplementary file 1, unless there are specific advantages to using different sets.

We address this in the revised submission by performing multiple hypothesis testing corrections. We utilize a permutation-based False Discovery Rate estimation, controlling the FDR at 15%. This results in a total of 55 genes as being identified as accelerated, and 1306 genes as decelerated. The supplementary tables in Supplementary file 1 have been modified to consistently utilize these numbers of accelerated and decelerated genes.

Related to the above, how does the RER method compare to the one used by Prudent et al.?

The two methods developed by Prudent et al. address the problem of identifying phenotypic-genomic associations in a different manner compared to our RER method. For one, our RER method relies on estimating sequence divergence by computing branch lengths on a Maximum Likelihood phylogenetic tree (number of substitutions per site). These calculations are based on standard models of amino acid or nucleotide evolution, whereas the methods in Prudent et al. rely on percent sequence identity and sequence mismatch to estimate the sequence divergence, which do not account for several confounding factors. For instance, sequence mismatch has a non-linear relationship with the actual divergence due to hidden mutation events such as back substitutions, multiple substitutions etc. This results in a consistent underestimation of actual divergence that further worsens as the divergence increases. Furthermore, sequence mismatch based measures fail to account for non-uniform substitution rates between nucleotide types and between sites [Li et al.]. Model-based measures of genetic divergence developed over the past years have been done so to control for many of these potentially confounding factors, and hence have been the standard methods employed in the field of phylogenetic inference [Kimura et al., Felsenstein et al., Cantor et al., Holmquist et al.].

Additionally, our normalization method handles missing orthologs for genes automatically by pruning away the corresponding branch(es) in all trees present, and subsequently generating relative rates, whereas it is unclear if the Prudent et al. methods handle such cases.

We performed a direct comparative analysis between our RER method and the GLS p-value method developed in Prudent et al. paper by studying the ability of the methods again to rank eye-specific regulatory regions around PAX6 on the same dataset. Our method performed better than the Prudent et al. method by putting the three known eye-specific enhancers at the top of the list of discovered regions (Author response image 1).

We additionally compared the performance of the methods in identifying high-ranking genes for enrichment in “eye morphology” and “eye physiology” as annotated by Mouse Genome Informatics. Using the same coding sequence alignments, our RER method enriched for known eye genes at a significantly higher rate than the Prudent et al. method (Author response image 2).

**Author response image 2. respfig2:** 

We believe that these analyses could be included in the manuscript if the reviewers deem it necessary, but we prefer the biological findings of the study to be the focus of the paper.

Figure 3 could additionally use as control a set of convergent genes in a different subset of species, such as the one reported by the authors for marine adaptations (Chikina et al., 2016).

We understand this would be one way to perform a control analysis but we believe that using four non-subterranean control species that have similar average genome-wide relative evolutionary rates to the subterranean mammals is perhaps a more conservative approach and strongly illustrates the distinction.

The supplemental excel tables in Supplementary file 1 could be better annotated, for example by adding legends at the top. Table S1 in Supplementary file 1 seems to be duplicated in the excel file.

We have added legends to the top of each supplementary table in Supplementary file 1, describing the columns present. We have additionally removed the duplication. E.g. for Table S1 “The genome-wide ranking of genes whose rates are positively associated with the subterranean branches was identified based on the Mann-Whitney U test p-values (column 2) that were significant at a FDR of 15% under a permutation-based FDR correction procedure (column 3). Other columns list brief gene descriptions (4), known biological function (5), tissue specificity if any (6), known disease links (7), the predominant evolutionary mode on subterranean branches as decided with codon models (8, see Materials and methods), and the subterranean species containing genetic lesions in the corresponding protein-coding sequences (9).”

Why does Table S7 in Supplementary file 1 only have 71 genes, given that the authors mention in the main text their using 98 eye developmental genes? It would be helpful to mark significantly accelerated genes in this table.

We started out with a set of 98 eye developmental genes and a filtering procedure removed a set of 27 genes as the branch lengths on their trees did not show sufficient variation (see Materials and methods). We have now rectified the mismatch in the numbers accordingly. “Using literature we also compiled a set of 71 important eye developmental genes”.

Wording in the subsection “Regressive evolution is limited to the lens, retina, and eye-specific developmental genes” is somewhat misleading and should be revised. ("we asked which eye tissues showed acceleration", whereas I think the authors are actually testing whether genes with eye-specific expression are enriched among accelerated genes).

We thank the reviewers for pointing out the distinction, and have suitably reworded the sentence: “we asked which genes with eye–tissue specific expressions showed acceleration and found that the cornea genes specifically expressed in cornea – a protective tissue of the outer eye – and the iris were not accelerated in subterranean species when compared to a set of randomly chosen genes”.

While the acceleration that the authors observe in Pax6 eye enhancers is consistent with relaxed constraint or "regression", in the absence of functional tests of these enhancers it is possible that the function of these enhancers is intact. In contrast, many of the vision genes that exhibit convergent sequence alteration have elevated dN/dS ratios or have stop codons and frame shift mutations that provide clear evidence of relaxed constraint. I don't feel that the authors need to perform functional tests of the Pax6 eye enhancers of subterranean species, but they should mention this caveat in the Discussion section.

We added this caveat as follows: “Although the strong rate acceleration in the three eye-specific enhancers of PAX6 suggests relaxation of constraint in the subterranean mammals, in the absence of functional tests we cannot be sure that the eye-specific activity is truly lost.”

In the Introduction, the authors cite Fang et al. as providing evidence for convergent evolution of circadian rhythm genes in blind mole-rat and naked mole-rat genomes. However, a corrigendum for this paper reported an error in the amino acid sequence alignment of the CLOCK genes. After correcting this error, there was no evidence of CLOCK protein sequence convergence between blink mole-rat and naked mole rat. Thus, it seems that Feng et al. may not provide support for the claim that circadian rhythm genes have cover gently evolved similar sequence changes. The authors should adjust their statements in the introduction accordingly. Fang et al., 2014. Nat Commun 5:1039-1052 for Corrigendum see http://www.nature.com/articles/ncomms9051

We thank the reviewers for pointing out the error and the reference to the corrigendum. We have now removed the corresponding incorrect statement from our Introduction section.

The authors repeatedly use the term "non-genic" to refer to non-coding sequences. The use of this term is somewhat ambiguous. For instance, the Pax6 gene has known (as well as putative) enhancer elements that are located within introns of the Elp4 gene. Thus, those elements lie within the Elp4 gene and could be considered "intra-genic" relative Elp4. The term "non-genic" should be replaced with the term "non-coding" throughout the manuscript. I also suggest that the authors replace "genic" with "coding," since the term "genic" can refer to the entire gene (not just coding exons, but also promoter regions, 5'UTR, 3'UTR, etc.).

We have replaced all occurrences of the term non-genic with non-coding. We agree that it makes the terminology more uniform and precise.

Gene and protein nomenclature: There are some errors and inconstancies in gene and protein names. For instance, "Lim2" is used to refer to the "LIM2" protein. "Pax6" is used instead of "PAX6." Also, when referring to "krt-/-" mutants, the first letter should be capitalized and the gene name should be italicized.

We have appropriately capitalized the protein names and the nomenclature regarding *Krt*-/- mutants.

Figure 3: The gene rank labeling in the histogram is truncated and needs to be extended to include the bars on the right end of the chart.

The bars corresponding to the ranks at the right end of the plot have now been added.

Figure 5: The abbreviation "Dev" and "Bgd" need to be defined in the figure legend.

We have added a legend describing all the abbreviations used in the plot.

Figure 6: The species that is used to diagram the gene region should be specified. If the species is mouse, then mouse gene nomenclature should be followed when labeling the diagram (first letter capitalized, name in italics).

The genomic window plotted in Figure 6 corresponds to the hg19 assembly and the labels defining the species and the chromosomal coordinates have been added to the plot.

Introduction includes long, highly detailed lists of adaptations – hard to follow and main points are lost. A figure or table summarising adaptations observed in subterranean taxa would be better.

We have now pruned out multiple paragraphs in the Introduction section, especially with regard to the various specific vision-related adaptations. We agree that it makes the Introduction more concise and gets to the objective of the study much quicker.

Writing is wordy and colloquial in many places. e.g. "Even though it is possible to visualize this accelerated rate of change in the Lim2 tree, it is more rigorous to use a quantitative framework."

We have reworded the particular sentence, and would appreciate further comments and feedback to strengthen the style of writing.

“To quantify this rate acceleration in the LIM2 tree, we normalized all branch lengths for the expected amount of change as defined by the genome-wide average divergence for each branch”.

Results include a lot of interpretation, sometimes quite speculative. As a result, much of the Discussion is repetitive.

We have pruned the Discussion to avoid repetition.

Overselling of enhancer results: I think the Pax6 enhancer results are a compelling example, suggesting regulatory elements may show similar convergent relaxed constraint to coding genes. However, title, Abstract, and Discussion suggest genome-wide analysis of enhancers was performed and found visual-specific enhancers accelerated ("successfully proved our hypothesis").

We have now included two large-scale validation analyses to expand from the example demonstration using Pax6 enhancers. We believe our new results clearly strengthen our argument for the hypothesis that convergent relaxation at the non-coding level can be used to identify vision-specific enhancers.

“Mole-accelerated non-coding elements are strongly enriched near transcription factors driving eye development”

“FANTOM5 eye enhancers show strong convergent acceleration in subterranean mammals”

Adaptation in tunneling-related genes: title and in the subsection “Skin-related genes were accelerated possibly in response to the demands of tunneling”. “reasons for their acceleration are split equally between relaxation of constraint and positive selection.” No numbers are given for # positively selected and total # tested – I believe only one positive selection example is listed (COL4A4). How is 'skin-related' defined – which GO terms?

We have now reworded the section describing the results pertaining to skin-related genes.

Significance of acceleration/deceleration.What is p-value threshold considered 'significant'?How was multiple testing accounted for?P values for top 1000 accelerated and decelerated in Supplementary file 1, Table S1/S3 very different – 614 accelerated have P <0.05; max for 1000 decelerated is P = 0.018.How does P value distribution compare to 2 control sets used for GO analysis? Or random sets of 4 species?

We address these questions in the revised submission by performing multiple hypothesis testing corrections. We utilize a permutation-based False Discovery Rate estimation, controlling the FDR at 15%. This results in a total of 55 genes as being identified as accelerated, and 1306 genes as decelerated.

The p-value distribution in random sets of 4 species was used for calculating the FDR q-values (see Materials and methods). At the same FDR of 15%, we have 10 genes significantly accelerated in control1 and 3 genes in control2. Unsurprisingly, given the small numbers of genes identified, neither of the control-accelerated gene sets showed any significantly enriched GO terms. Given the highly similar nature of results across the two controls, we decided to report the findings based on just one of the sets of control species (control 1 – pika, guinea pig, squirrel, cow) in the revised submission.

The numbers of genes identified as having decelerated rates in the moles is 1306, at a FDR of 15%. At the same FDR, 626 genes are decelerated in the control species. Mole-decelerated genes despite being 20-fold more in number than mole-accelerated genes, do not show significant functional enrichment. We find only 1 category as significantly enriched, whereas 24 GO term categories are significantly enriched across control species. We are separately investigating the wide differences in the numbers of genes that are being detected as accelerated versus decelerated alongside other potential methodological refinements, and focus the results of this paper on the biological findings.

Acceleration/ DecelerationForeground speciesNumber of significant genes at FDR of 15%Number of significant GO termsAccelerationMoles5515Control100DecelerationMoles13061Control62624

Rods vs. cone genes analysis (subsection “Each subterranean species exhibited selective gene acceleration consistent with its visual capacities”)It is hard to evaluate these results because I can't find a legend explaining ++s and --s in Supplementary file 1, Table S8 (top) and there are several empty cells in the pseudogene section (bottom). Nevertheless, this section is overly descriptive and I don't think much can be concluded based on inconsistent patterns across 3 genes/category.

Upon consideration of reviewers’ comments’, we have now removed the results of the section corresponding to the rods vs. cone genes analysis. We feel that with the new results corresponding to the non-coding elements, the results of the rod/cone genes analyses perhaps become less significant.

Some Methods are unclear:Beginning of Materials and methods should clearly explain overall rationale for data included in RER. e.g., describe 100-spp alignment, criteria for inclusion in proteome average tree, minimum # of subterranean sequences required for analysis (vs. 10 overall).

We have now addressed these points in more detail in the paper.

“Using the 100-way 100 vertebrate species amino acid alignments from the multiz alignment available at the UCSC genome browser [Blanchette et al., 2004; Harris, 2007], those alignments with a minimum of 10 species were selected for study, additionally requiring the presence of at least two subterranean species. We pruned each alignment to include 39 species of interest represented in the proteome-wide average tree (Figure 1) after adding the BMR ortholog of the corresponding gene sequence to this alignment as described in the previous section”.

Figure 3 – how was this analysis done? What counts as 'mole acceleration' and 'visual perception'?

We have now added the appropriate definitions.

“False Discovery Rate analysis was performed on probabilities resulting from the Mann-Whitney U test. We employ an empirical permutation-based FDR calculation, often the standard approach in genome-wide analysis. We generated 10,000 null datasets, where each dataset was obtained by randomly permuting the species labels of the relative rates. This process is equivalent to calculating the Mann-Whitney U test and correlation analysis over four random branches vs the rest instead of over the binary variable of “subterranean” vs. “aboveground” branches. The subsequent permuted datasets were used to estimate the FDR q-values for each p-value in our subterranean correlation analysis. Genes showing a correlation greater than equal to 0.5 in the Mann-Whitney U test and significant at a FDR of 15% are considered mole-accelerated genes, and the p-value reflecting the strength of this acceleration is referred to as mole-acceleration.”

Arbitrary/inconsistent gene list #s. Why top 30 genes in Table 1? Top 500 accelerated in GO analysis Table 2? Why 1000 top genes included in supplementary tables in Supplementary file 1? Why 200 included in PAML analysis?

We have homogenized the gene sets used across various analysis.

Pax6 enhancer analysis – (subsection “Eye-specific enhancers of PAX6 show convergent acceleration in subterranean mammals”, second paragraph), if enhancers in 200 kb window, why 500 kb 'window of interest', is 200 kb in center? Specify locations.

We performed the scan across a larger window to include a larger number of conserved non-coding elements and potentially discover new elements that may be vision-specific. Labels corresponding to the genomic locations have now been added to Figure 6.

Tissue-specific expression (subsection “Regressive evolution is limited to the lens, retina, and eye-specific developmental genes”). It is unclear what 'significant differential expression only in the tissue of interest' means – significantly higher expression only in one tissue compared to background?

The genes had significantly higher expression in that tissue compared to all the other tissues.

Figure 1 – include scientific names.

We decided against including scientific names as it makes the text in the figure too cramped.

Figure 1, Figure 2, Figure 4: what are unlabeled blue dots at bottom? Why are species ordered alphabetically by common name (it appears, not specified) instead of by phylogeny?

The unlabeled blue dots correspond to ancestral branches in the tree. We now explain this in the figure legend. The dots are not ordered by phylogeny because the relative rates are obtained after correcting for the inherent phylogenetic relationship between the branch lengths. The alphabetical order was used merely for convenience.

Figure 5 – r2? P value?

Added.

Figure 5 – not described in legend; shades hard to distinguish.

Added.

Figure 6 – indicate 200 kb and 500 kb windows for analysis.

Added the legend corresponding to the chromosomal coordinates. (Refer our twenty-fourth response to point 1).

Subsection “Most subterranean-accelerated genes are under relaxed constraint” – how many mole-accelerated genes show lesions in 1 or more subterranean species?Subsection “Regressive evolution is limited to the lens, retina, and eye-specific developmental genes”, “clustered at the top of the accelerated list” – what does this mean? Overall list in Supplementary file 1, Table S1?

We have replaced the word ‘clustered’ with ‘present’ in the sentence now. The list here refers to the list of mole-acceleration of developmental genes.

2) Extending Pax6 enhancer analysis (specific details)The analysis on non-coding regions (Figure 6) must be expanded. If the point the authors want to make is that their approach is applicable to non-coding regions, this approach should be applied to all conserved non-coding elements rather than a single locus. An increased scope of this analysis would add novelty to the author's findings. Can one still detect an enrichment of eye-related gene annotations proximal to enhancers with accelerated evolution in subterranean mammals? How does it compare with that of coding regions?

We have now added two new sections to the paper expanding from the example analysis using Pax6 enhancers. We have analyzed a total of roughly 16,000 non-coding elements, 8,000 elements scanned for the study contrasting two sets of transcription factors and 8,000 elements corresponding to FANTOM5 putative enhancer elements. We believe these new results further strengthen our argument that convergent rate acceleration can identify candidate vision-specific genetic elements in the mammalian genome.

The authors highlight their search for convergent patterns of sequence divergence as a means to identify genetic elements that may underlie changes in phenotype. If the authors analyzed each subterranean mammal on its own (looking at sequence divergence but not convergence between subterranean species), do they see strong enrichment of vision-related GO terms in each species? Do they also see other non-vision GO terms (e.g., hypoxia)? How different are the GO enrichments for accelerated proteins in individual subterranean species vs. the list of proteins that show convergent accelerations among subterranean species?

We have considered the question of looking at relative rates in subterranean mammals individually as against convergent rate changes. Generally, we do not see any GO term significantly enriched from these individual analyses, none related to vision at any rate. Due to the very weak nature of the associations that come out of these analyses, we do not include these results in the main manuscript.

3) Other (non-required) pointsAs a strategy to inform the molecular basis of phenotypic traits, the author's approach is an alternative to direct experimental mapping across mammals (e.g. of gene expression; Ma et al., 2016). The authors hypothesize that genomic regions under accelerated evolution in subterranean mammals may inform candidate loci for congenital eye disorders, but no direct evidence is provided. The manuscript would tremendously benefit from substantiating this hypothesis with experimental data, for example by evaluating a few of the accelerated loci (genes or regulatory regions) in animal models of eye development, such as mouse or zebrafish. Can the authors show a requirement for some of their novel candidate elements in eye development or eye-specific expression? (Please see (Kvon et al., 2016) for a similar approach).

We thank the reviewers for suggesting a complementary strategy to our approach. We are currently in the process of conducting experiments that can provide direct evidence in validating our candidate vision-specific genetic elements. However, because we are developing an expression construct system for those experiments, they are more than a year from being completed. We hope that these experiments will be the subject of future publications.